# Estimating the protein burden limit of yeast cells by measuring the expression limits of glycolytic proteins

Yuichi Eguchi[1], Koji Makanae[2], Tomohisa Hasunuma[3], Yuko Ishibashi[4], Keiji Kito[4], Hisao Moriya[1,2]*

[1]Graduate School of Environmental and Life Science, Okayama University, Okayama, Japan; [2]Research Core for Interdisciplinary Sciences, Okayama University, Okayama, Japan; [3]Graduate School of Science, Technology and Innovation, Kobe University, Kobe, Japan; [4]Department of Life Sciences, School of Agriculture, Meiji University, Kawasaki, Japan

**Abstract** The ultimate overexpression of a protein could cause growth defects, which are known as the protein burden. However, the expression limit at which the protein-burden effect is triggered is still unclear. To estimate this limit, we systematically measured the overexpression limits of glycolytic proteins in *Saccharomyces cerevisiae*. The limits of some glycolytic proteins were up to 15% of the total cellular protein. These limits were independent of the proteins' catalytic activities, a finding that was supported by an in silico analysis. Some proteins had low expression limits that were explained by their localization and metabolic perturbations. The codon usage should be highly optimized to trigger the protein-burden effect, even under strong transcriptional induction. The S–S-bond-connected aggregation mediated by the cysteine residues of a protein might affect its expression limit. Theoretically, only non-harmful proteins could be expressed up to the protein-burden limit. Therefore, we established a framework to distinguish proteins that are harmful and non-harmful upon overexpression.
DOI: https://doi.org/10.7554/eLife.34595.001

*For correspondence:
hisaom@cc.okayama-u.ac.jp

Competing interests: The authors declare that no competing interests exist.

## Introduction

Protein overexpression is sometimes harmful to cellular growth (*Makanae et al., 2013*; *Sopko et al., 2006*), and a few mechanisms that could result in overexpression-triggered growth defects have been proposed (*Moriya, 2015*). Resource overload, stoichiometric imbalance, promiscuous interaction, and pathway modulation are triggered upon overexpression of, respectively, (i) a protein that has a high demand of cellular resources (*Dong et al., 1995*; *Kintaka et al., 2016*; *Stoebel et al., 2008*), (ii) a protein that is part of a protein complex (*Kaizu et al., 2010*; *Makanae et al., 2013*; *Papp et al., 2003*), (iii) a protein with a nonspecific interaction domain (*Ma et al., 2010*; *Vavouri et al., 2009*), and (iv) a protein that catalyzes a pathway (*Prelich, 2012*; *Youn et al., 2017*). The mechanism of protein overexpression-triggered growth defects depends on the protein's structural and functional characteristics, which are not always fully understood yet. Therefore, it is still difficult to predict whether the overexpression of a particular protein will be harmful to cellular growth and which mechanisms cause the harmful effect.

The ultimate overexpression of a protein could be harmful for cellular growth, because it monopolizes and depletes limited resources that are involved in protein production, such as ribosomes and aminoacyl-tRNAs (*Gong et al., 2006*; *Shachrai et al., 2010*; *Vind et al., 1993*). This phenomenon is known as the protein burden/cost effect (*Kafri et al., 2016*; *Snoep et al., 1995*). Proteins that have no harmful effects on cellular functions can be overexpressed up to a level that causes protein-

**eLife digest** If a cell makes too much of a given protein, it can sometimes cause problems and impair the cell's growth. Overproducing some proteins may deplete the cell's limited resources, meaning it does not have enough to make other more essential proteins. This phenomenon is known as the protein burden effect. Theoretically, only harmless proteins can be overproduced up to a level where growth would be impaired in this way. Conversely, if an overproduced protein causes harm before it becomes a burden on resources, scientists must consider other mechanisms to explain the cell's problems, namely that the protein itself is harmful.

Knowing the ultimate level of protein production that could cause the protein burden effect – the protein burden limit – would allow scientists to distinguish between harmful and non-harmful proteins. However, to date, this limit had not been defined for any cell.

Eguchi et al. have now tried to estimate the protein burden limit for budding yeast – one of the best-studied experimental organisms. The experiments first focused on enzymes involved in alcoholic fermentation because they were expected to be non-harmful. Some of these enzymes were overproduced to the level were the made up 15% of all the cell's proteins before they started to cause growth defects. The same results were seen with versions of the enzymes that had been mutated to be less active, leading Eguchi et al. to conclude that this level is the protein burden limit.

In other experiments, harmful enzymes could only be overproduced to levels that were far less than this proposed protein burden limit. These enzymes caused problems for the yeast in several ways, including interfering with biochemical reactions and forming large aggregates in the cell. Lastly, Eguchi et al. looked at the yeast's genetic code and saw that most of its genes seemed to have evolved to specifically limit the production of proteins to a level that would avoid the unwanted protein burden effect.

Together these findings establish a framework to clearly distinguish between harmful and non-harmful proteins. This framework will be useful to understand the different reasons why the overproduction of certain proteins, which is seen in neurodegenerative diseases and cancer cells, can cause problems for cells.

DOI: https://doi.org/10.7554/eLife.34595.002

burden–triggered growth defects. Conversely, if a protein cannot be overexpressed up to that level because it adversely affects cellular functioning, then overexpression of that protein will cause growth defects at relatively low expression levels, and we should consider mechanisms causing the defects.

We previously developed a genetic tug-of-war (gTOW) method that can be used to estimate the expression limit of a target protein that triggers growth defects in the yeast *Saccharomyces cerevisiae* (*Makanae et al., 2013*; *Moriya et al., 2006 , 2012* ). We estimated that the expression limit of a green fluorescent protein (GFP) was about 15% of the total cellular protein in *S. cerevisiae* (*Kintaka et al., 2016*). Because GFP is a highly structured cytoplasmic protein unrelated to the cellular functions of yeast and thus harmless, this level could be considered the expression limit for any protein that causes growth defects triggered by the protein-burden effect.

We predicted that the expression limits of some native highly expressed glycolytic proteins would be similar (>15% of the total cellular protein) (*Moriya, 2015*), suggesting that overexpression of these proteins would be harmless even though they have metabolic functions in yeast. The prediction was performed by the calculation of the proteins' native expression levels (*Kulak et al., 2014*) and their gene copy number limits as determined by gTOW analysis, and has not yet been experimentally validated (*Moriya, 2015*). In this study, therefore, we tried to measure the expression limits of 29 glycolytic proteins to assess whether they are expressed up to levels that cause growth defects triggered by the protein-burden effect. There are five reasons why we chose glycolytic proteins: (1) because they are generally highly expressed and thus considered non-harmful upon high-level expression, they are excellent targets for examining whether they are expressed up to the protein-burden limit; (2) because they have been intensely studied, we have information that can allow us to manipulate their catalytic activities; (3) because the glycolytic pathway is one of the best-known metabolic pathways, we can predict and measure metabolic changes upon overexpression of

these proteins; (4) they include a heteromer (Pfks), a mitochondrially localized protein (Adh3), and membrane proteins (Hxts), so we can assess how these characteristics affect expression limits; and (5) they include paralogs whose expressions are differently regulated, so that we can test how their differences affect their expression limits.

We found that the expression limits of most of the 29 proteins were comparable to that of GFP and were independently determined by their catalytic activities, as suggested by a kinetic model of yeast glycolysis, confirming that their overexpression was harmless. Also, some of the proteins had far lower expression limits than those that would create a protein burden (the expression limit of GFP), and their harmful effects were derived from their localization and metabolic perturbations. Owing to their codon optimality, native poorly expressed isozymes were not produced at levels sufficient to cause growth defects, even when they were expressed from the strong *TDH3* promoter on the multicopy plasmids. Some glycolytic proteins formed S–S-bond-mediated aggregates when overexpressed, and this aggregation also seemed to restrict their expression limits.

## Results

### Overexpression of glycolytic proteins from a strong promoter on a multicopy plasmid causes growth defects

*Figure 1A* shows the experimental system (plasmid) used to express glycolytic proteins to limits that cause growth defects. The target glycolytic proteins analyzed in this study and their characteristics are summarized in *Supplementary file 1*. We cloned each target gene on the gTOW plasmid (pTOW40836) (*Moriya et al., 2012*), such that the target protein was expressed under the control of the strong *TDH3* promoter. *PFK1* and *PFK2* were exceptionally expressed from the less-strong *PYK1/CDC19* promoter because their expression from the *TDH3* promoter was too strong and consequently the growth of the transformants was very poor (data not shown). The plasmids were used to transform the *S. cerevisiae* strain BY4741 (*ura3Δ leu2Δ*). Copy numbers of the plasmid within the cell were controlled by changing growth conditions: up to 35 copies per cell in +leucine (–uracil) conditions (low-copy conditions) and up to 150 copies per cell in –leucine conditions (high-copy conditions) due to the biases *2 μm ORI* and *leu2-89* (*Moriya et al., 2012*). In this experimental system, the maximum growth rates of the cells with the vector in +leucine conditions are much greater than those in –leucine conditions (see *Figure 1B–C*), probably because the copy number of *leu2-89* is not sufficient tosupport fully the leucine requirement in –leucine conditions. We measured the expression limits of most of the 29 target proteins in low-copy conditions because the expression levels produced under these conditions were already sufficient to cause growth defects.

We first measured the growth rates of cells harboring gTOW plasmids. As shown in *Figure 1B*, all cells expressing glycolytic proteins, with the exceptions of those expressing *HXT1*, *HXT3*, and *HXT4*, showed significant growth retardation compared to the vector control cells in low-copy conditions (p<0.01, Welch's *t*-test, *Figure 1—source data 1*), indicating that expression of most of the glycolytic proteins caused growth defects. This observation was confirmed by the growth measurement in the high-copy conditions shown in *Figure 1C*, as cells expressing most of the glycolytic proteins did not grow in these conditions. Cells expressing *GLK1*, *FBA1*, *GPM1*, *PYK2*, *PDC6*, *ADH5*, and *ADH4* could grow in high-copy conditions, although their growth rate was significantly lower than that of the vector control (p<0.01, Welch's *t*-test, *Figure 1—source data 1*).

As previously reported, the copy number of the gTOW plasmid inside the cell inversely reflects the deleterious effect of protein expression from the plasmid due to the gTOW effect: the plasmid copy number is low if expression of the target protein is harmful to cellular growth, and high if expression of the target protein is less harmful (*Kintaka et al., 2016*; *Makanae et al., 2013*; *Moriya et al., 2006, 2012*). *Figure 1D and E* show the copy numbers of gTOW plasmids in low- and high-copy conditions. In high-copy conditions, the copy numbers of gTOW plasmids expressing only *GLK1*, *FBA1*, *GPM1*, *PYK2*, *PDC6*, and *ADH4* were determined because yeast containing plasmids expressing the other protein-coding genes failed to grow. The copy numbers of all gTOW plasmids containing target genes were significantly lower than those containing the empty vector (p<0.05, Welch's *t*-test, *Figure 1—source data 1*), confirming that they were expressed up to levels that caused growth defects in this experimental system. Because the copy numbers of plasmids expressing most of the glycolytic proteins tested here, other than *GLK1*, *PYK2*, and *ADH4*,

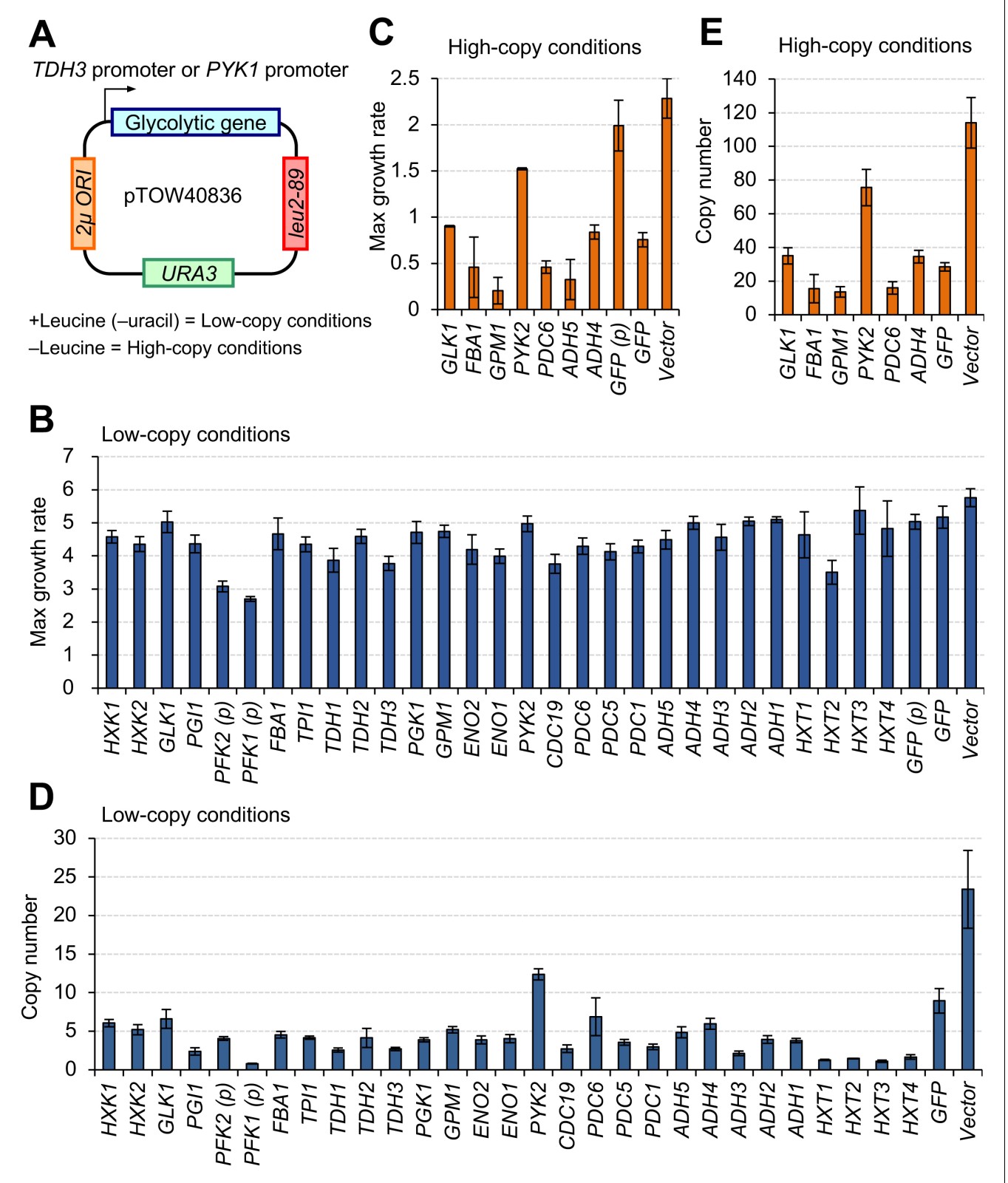

**Figure 1.** Overexpression of most glycolytic proteins using a strong promoter and a multicopy plasmid causes growth defects. (**A**) The plasmid used in this study. Each glycolytic gene was cloned into a 2-µm-based multicopy plasmid (pTOW40836) and expressed from the *TDH3* promoter (*TDH3pro*) (with the exception of *PYK1* which was expressed from the *PYK1* promoter [*PYK1pro*], represented in the figure as (*p*)). In +leucine conditions, the copy number of the plasmid is relatively low (~30). In −leucine conditions, the copy number goes up to 150 copies per cell due to the bias of *leu2-89*

*Figure 1 continued on next page*

Figure 1 continued

(*Moriya et al., 2012*). Here, we designate these conditions low- and high-copy conditions, respectively. (**B** and **C**) Maximum growth rate of yeast cells harboring the plasmid overexpressing each glycolytic protein in the indicated growth conditions. The unit is $min^{-1} \times 10^{-4}$. (**D** and **E**) Copy number of the plasmid overexpressing each glycolytic protein in the indicated growth conditions. The unit is copy number per haploid genome. The error bars shows the standard deviation of at least three independent biological measurements.

DOI: https://doi.org/10.7554/eLife.34595.003

The following source data is available for figure 1:

**Source data 1.** This spreadsheet contains all data and statistical values associated with the figure.

DOI: https://doi.org/10.7554/eLife.34595.004

were not greater than the copy number of GFP, the expression of most glycolytic proteins in this experimental system seemed no less defective than that of GFP.

We concluded that most glycolytic proteins were expressed close to their upper limits, even in low-copy conditions, and that a copy number increase in high-copy conditions was required to express *GLK1*, *FBA1*, *GPM1*, *PYK2*, *PDC6*, and *ADH4* to their limits.

## Measurement of the expression limits of glycolytic proteins

Next, we measured the expression levels of proteins within the cells, overexpressing them from the gTOW plasmid. *Figure 2A* shows how protein abundance was estimated. As reported previously, when GFP is expressed up to its expression limit (and probably to the level required to trigger the protein-burden effect), the protein is visible within whole cellular proteins separated by sodium dodecyl sulfate polyacrylamide gel electrophoresis (SDS-PAGE) (*Kintaka et al., 2016*). Because most glycolytic proteins were also expressed to similar levels by our experimental system, we measured the expression levels in arbitrary units (AU) as the relative intensities of target protein bands within the total protein separated by SDS-PAGE. The AU is considered to reflect the total number of amino acids within the band, and the relative number of protein molecules can be estimated by dividing AU by the protein length. When two proteins of different sizes give bands of the same number of AUs, the molecule number of the larger protein in the band should be lower than that of the smaller protein. The relationship between the AU and the percentage of total protein that we previously reported (*Kintaka et al., 2016*) was estimated, as shown in *Figure 2—figure supplement 1*, as *% total protein = 5.5 ×* AU. Representative images of SDS-PAGE-separated total proteins from cells harboring gTOW plasmids containing the target glycolytic protein genes are shown in *Figure 2—figure supplement 2*. As shown in *Figure 2B*, most proteins were expressed at levels high enough to make them visible within the SDS-PAGE–separated whole cellular proteins, and the expression levels of Pgk1, Gmp1, Eno2, and Eno1 were higher than that of GFP. By contrast, the expression levels of Pfk1, Adh3, and Hxts were almost undetectable with this experimental system. The *x*-fold increase in the expression of each target protein over its native level is shown in *Figure 2—figure supplement 3*. The expression of some proteins was increased more than 10,000-fold in this experimental system. The expression of Tdh3 and Gpm1 further increased in –leucine conditions (*Figure 2C*), but the cells in this condition had stunted growth (*Figure 1C*).

There could be two reasons that explain why the expression level of a protein is low: (i) its strong overexpression is harmful to cellular growth and (ii) its expression is repressed. We can distinguish these two possibilities by comparing the copy numbers and protein abundance as shown in *Figure 2D* because the copy number of the plasmid inversely reflects the deflective effect of protein expression as described above. Overexpression of Pfk1, Adh3, and Hxts seemed harmful because their copy numbers were lower than the those of the other proteins (red circles in *Figure 2D*). By contrast, the expression of Glk1, Pyk2, and Pdc6 seemed to be repressed because their copy numbers were higher than those of the other proteins (blue circles in *Figure 2D*). The relationship between protein expression levels and copy numbers (shown in *Figure 2D*) also suggested that expression levels are not solely determined by the promoter, because there was no significant correlation between the expression levels and plasmid copy numbers (Pearson $r=0.28$, $p=0.12$).

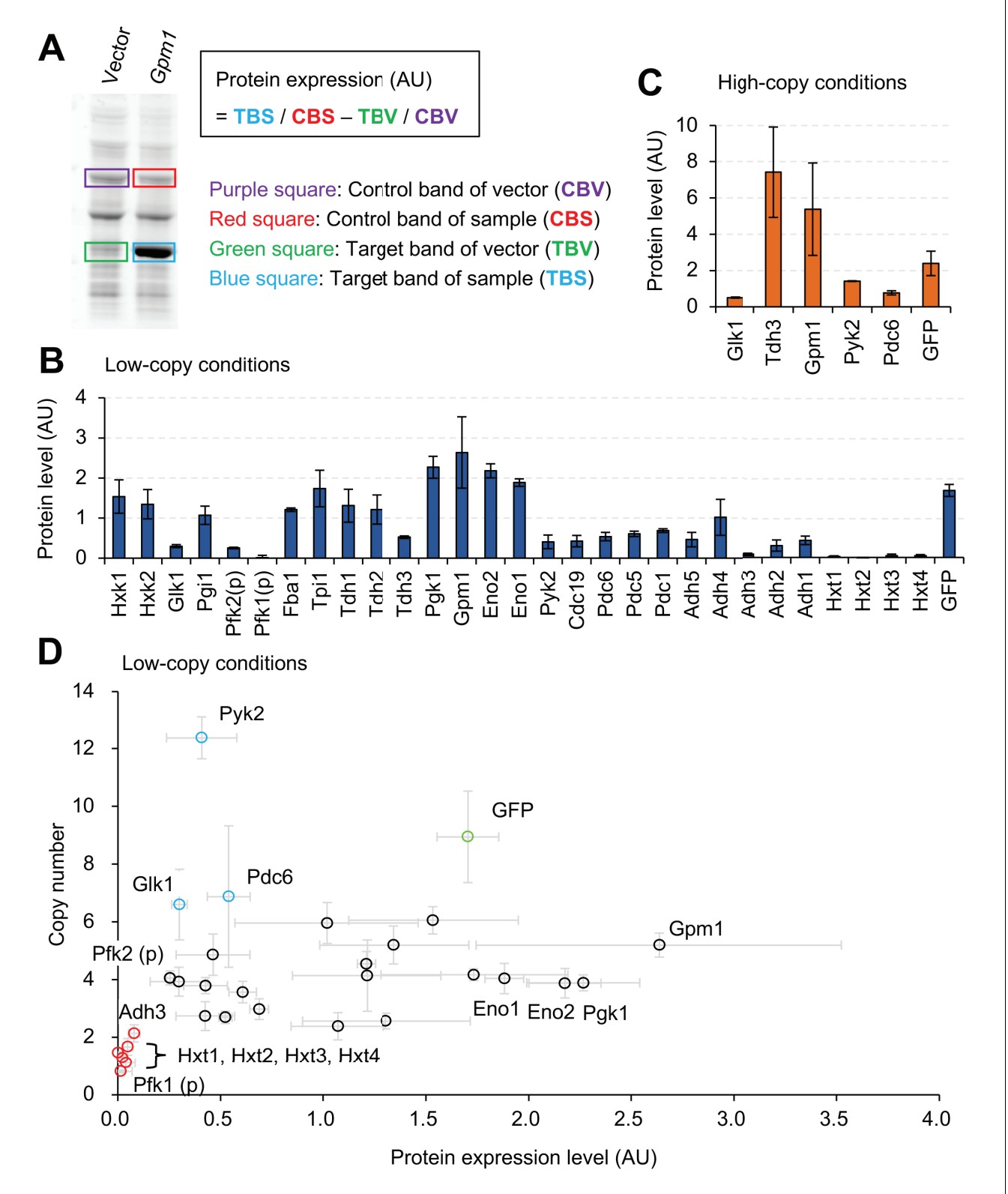

**Figure 2.** Expression limits of glycolytic proteins. (**A**) Measurement of the expression level of an overexpressed glycolytic protein. Whole cellular proteins were stained by a fluorescent dye and separated by SDS-PAGE. The overexpression of the target protein (TBS), estimated from the intensity of it's band on the gel an its molecular weight, was compared with the corresponding expression of the vector control (TBV), after normalization using control bands (CBS and CBV) to calculate protein expression (AU). Measurement of Gpm1 expression level is shown as an example. (**B** and **C**) The

*Figure 2 continued*

expression level of glycolytic proteins overexpressed using the experimental system shown in *Figure 1* in the indicated conditions. The *TDH3* promoter was used for the expression of all genes except where (p) indicates the use of the *PYK1* promoter. (D) Relationship between copy number and protein level in low-copy conditions. The copy number data are the same as in *Figure 1D*, and the protein level data are the same as in *Figure 2B*. The error bars shows the standard deviation of at least three independent biological measurements.

DOI: https://doi.org/10.7554/eLife.34595.005

The following source data and figure supplements are available for figure 2:

**Source data 1.** This spreadsheet contains all data and statistical values associated with the figure.
DOI: https://doi.org/10.7554/eLife.34595.009
**Figure supplement 1.** Relationship between percentage of total protein in (*Kintaka et al., 2016*) and protein level (AU) in this study.
DOI: https://doi.org/10.7554/eLife.34595.006
**Figure supplement 2.** SDS-PAGE-separated total cellular proteins of the cells overexpressing glycolytic proteins.
DOI: https://doi.org/10.7554/eLife.34595.007
**Figure supplement 3.** Estimated fold overexpression of glycolytic proteins in this study.
DOI: https://doi.org/10.7554/eLife.34595.008

## Mutations in catalytic centers do not affect the expression limits of most glycolytic proteins

Next, we tried to reveal the factors causing harmful effects that restrict expression limits, and the mechanisms that repress protein expression. Overproduction of glycolytic proteins might cause metabolic perturbations by accelerating the reactions that these proteins catalyze. To test whether growth inhibition caused by metabolic perturbations limits the expression levels of glycolytic proteins, we analyzed the expression limits of mutant proteins with reduced enzymatic activities by introducing mutations into the catalytic centers (here we call the mutant a 'CC mutant'). The mutations introduced into the glycolytic proteins are summarized in *Supplementary file 1*. *Figure 3A* shows the expression levels of wild-type and mutant proteins in low-copy conditions. The expression levels of all proteins except Pfk1, Fba1, Tdh3, and Eno1 were not significantly changed by introducing mutations. The expression levels of mutant Pfk1 and Tdh3 were significantly higher than those of wild-type proteins, and the expression levels of mutant Fba1 and Pgk1were significantly lower than those of wild-type proteins ($p < 0.05$, Welch's *t*-test, *Figure 3—source data 1*). For Pfk1, Tdh3, and Pfk2 (which catalyzes the same reaction with Pfk1), we further analyzed the expression levels in high-copy conditions (*Figure 3B*). The expression level of mutant Pfk2 significantly increased when compared with that of wild-type Pfk2 ($p = 0.046$, Welch's *t*-test). The expression levels of both wild-type and mutant Pfk1 were almost undetectable in these conditions, probably because their high-level expression was too toxic to the yeast cells. Because the expression level of wild-type Tdh3 was greater than that of mutant Tdh3, the enzymatic activity of Tdh3 probably did not restrict its protein expression limit. We concluded that the expression limits of most of the glycolytic proteins studied here are not restricted by metabolic perturbations triggered by their overproduction, whereas the expression limits of Pfk1 and Pfk2 are exceptionally restricted by metabolic perturbations. The expression levels of mutant Pfk1 and Pfk2, however, remained markedly lower than those of other glycolytic proteins, suggesting that other factors also influence their expression limits.

## Mitochondrial localization restricts the expression limit of Adh3

Next, we focused on Adh3, whose expression level was lower than those of the other glycolytic proteins, probably because high-level expression of Adh3 is harmful (*Figure 2D*). This harmful effect, however, is not triggered by metabolic perturbations, because the expression level of the mutant Adh3 with reduced enzymatic activity was almost the same as that of wild-type Adh3 (*Figure 3A*). Among the glycolytic proteins tested in this study, Adh3 alone is a mitochondrial protein (*Young and Pilgrim, 1985*). To test whether the mitochondrial localization of Adh3 restricts its protein expression limit, we constructed a mutant without the mitochondrial targeting sequence (ΔMTS-Adh3, *Figure 3—figure supplement 1*) and compared its expression level to that of wild-type Adh3. As shown in *Figure 3C and D*, the expression level of ΔMTS-Adh3 was about three times higher than that of wild-type Adh3. We concluded that the mitochondrial localization of Adh3 restricts its expression limit, probably because the high-level expression of this mitochondrial protein causes growth defects due to overloading of mitochondrial transport resources (*Kintaka et al., 2016*).

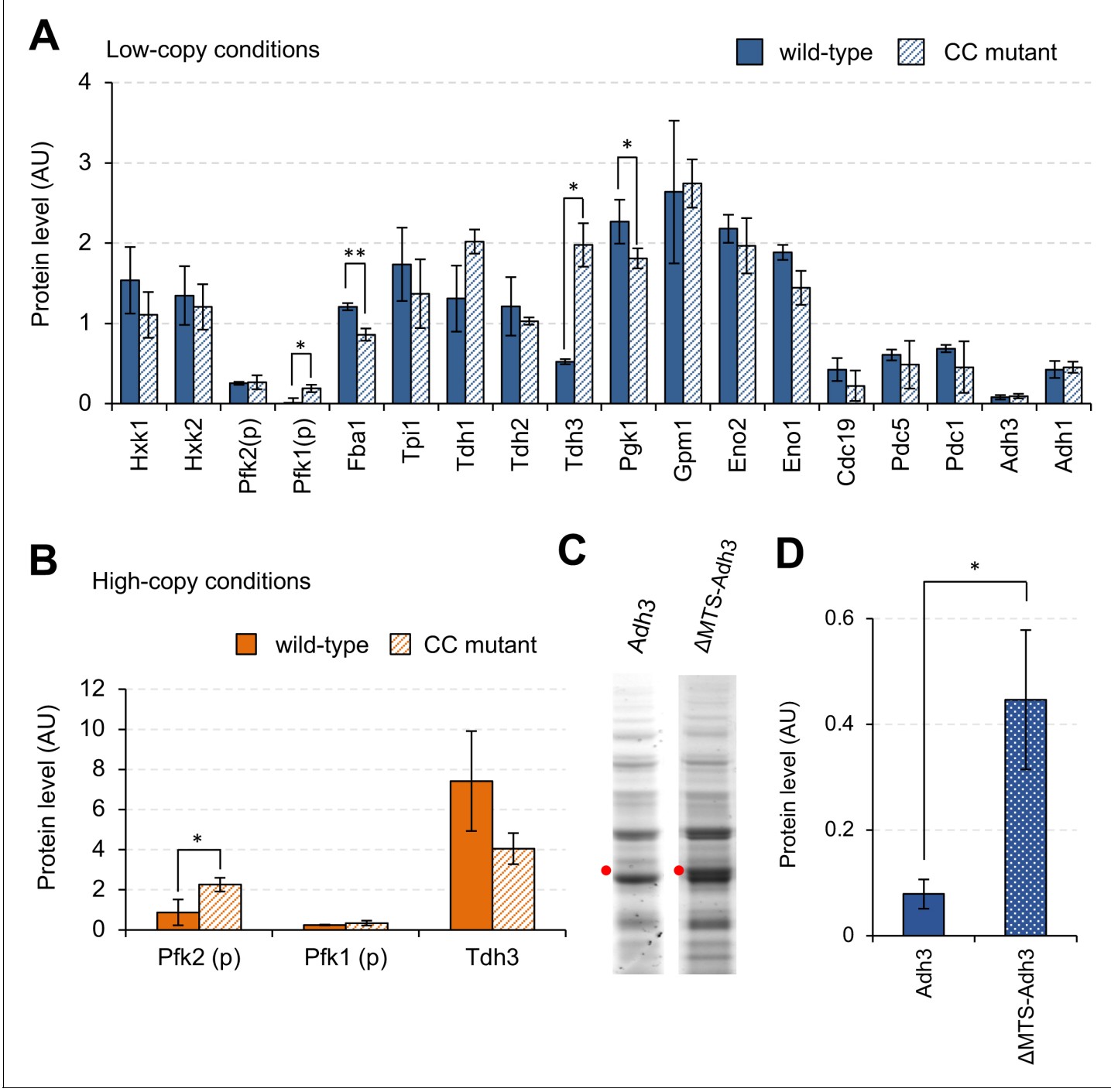

**Figure 3.** Effects of mutations on the expression limits of glycolytic proteins. (**A** and **B**) Expression levels of wild-type and CC mutant glycolytic proteins in the indicated conditions. Each CC mutant has a mutation in the position shown in *Supplementary file 1*. (**C**) SDS-PAGE gel images of whole cellular proteins overexpressing Adh3 and ΔMTS-Adh3 in low-copy conditions. Red dots indicate the expected sizes of the target proteins. (**D**) Protein expression levels of Adh3 and ΔMTS-Adh3 in low-copy conditions. The error bars indicates the standard deviation of the mean. *$p<0.05$; **$p<0.01$ in Welch's *t*-test.

DOI: https://doi.org/10.7554/eLife.34595.010

The following source data and figure supplement are available for figure 3:

**Source data 1.** This spreadsheet contains all data and statistical values associated with the figure.

DOI: https://doi.org/10.7554/eLife.34595.012

*Figure 3 continued on next page*

*Figure 3 continued*

**Figure supplement 1.** Comparison of the N-terminal amino-acid sequences of Adh1, Adn5, Adh3, and Adh3 mutants without the mitochondrial targeting sequence (ΔMTS-Adh3).

DOI: https://doi.org/10.7554/eLife.34595.011

## Metabolic perturbations triggered upon overexpression of glycolytic proteins

The results suggested that the overexpression of most glycolytic proteins do not cause serious metabolic perturbations. To test whether this speculation is theoretically supported, we used a kinetic model of the yeast glycolytic pathway (*Smallbone et al., 2013*); a schematic diagram of which is shown in *Figure 4—figure supplement 1*. *Figure 4A–D* shows the *x*-fold change of glycolytic metabolites in simulations in which each glycolytic protein is overexpressed up to 128-fold compared with the wild-type simulation. Overproduction of 14 of 20 glycolytic proteins did not cause more than a two-fold metabolic change (gray lines in *Figure 4A*), indicating that overexpression of most glycolytic proteins does not cause serious metabolic perturbations. By contrast, the overproduction of Hxk1 and Hxk2 affected glycolytic metabolism throughout, and the overproduction of Pdc1 and Cdc19 affected metabolism locally (*Figure 4B–C*). Because the experimental results using CC mutants suggested that their overexpression did not trigger metabolic perturbations leading to the growth defects, unknown mechanisms to explain the discrepancy might exist. Overproduction of Pfk1 or Pfk2 did not cause a metabolic change, because, in the model, these individual enzymes did not catalyze the Pfk reaction whereas the Pfk1–Pfk2 complex did. Simultaneous overproduction of both Pfk1 and Pfk2 caused severe metabolic changes (*Figure 4D*), whose pattern was quite similar to the changes caused by Hxk1 and Hxk2 overexpression (*Figure 4—source data 1*) (except G6P and F6P levels did not change as these metabolites are upstream of the Pfk reaction). Although metabolic changes upon overexpression of Pfks and Hxks showed a similar pattern, overexpression of Pfks but not Hxks caused growth defects (*Figures 1* and *2*), and catalytic mutations of only Pfks increased the expression limit of this protein (*Figure 3*). Hence, the metabolic changes observed in the simulation do not by themselves explain the growth defects triggered by the overexpression of Pfks.

To further characterize physiological conditions that are triggered by the overexpression of Pfks, we next analyzed metabolic changes in yeast cells overexpressing wild-type and CC mutant Pfk2 over the vector control by measuring 35 metabolites (*Figure 5—source data 1*), because the CC mutants showed increased expression limits (*Figure 3B*). *Figure 5A* shows changes in the levels of nine glycolytic metabolites. Overexpression of both wild-type and CC mutant Pfk2 triggered significant reductions in some metabolites (p<0.05, Welch's *t*-test, *Figure 5—source data 1*). Moreover, the patterns of metabolic changes were inconsistent with those predicted by the model (*Figure 5—figure supplement 1*). These metabolic reductions were thus not triggered by the catalytic activity of Pfk2. We noticed, however, that the level of F16bP in the cells overexpressing wild-type Pfk2 was >3-fold higher than that in the CC mutant Pfk2 (*Figure 5A*, p<0.05, Welch's *t*-test, *Figure 5—source data 1*). F16bP is the product of Pfk catalysis and the simulation predicted an increase in the F16bP level upon overexpression of Pfks (*Figure 4D*), suggesting that the catalytic activity of Pfk2 triggers this metabolic difference.

We next measured metabolic changes in 29 metabolites in cells overexpressing wild-type Pfk1 and Tdh3 and their CC mutants because these CC mutants also showed increased expression limits (*Figure 3A*). As shown in *Figure 5B and C*, levels of glycolytic metabolites in the cells overexpressing wild-type Pfk1 and Tdh3 were not changed more than three-fold over the vector control. We did not observe any reproducible increase in F16bP level in the cells overexpressing wild-type Pfk1 over levels in its CC mutant. Moreover, overall metabolic changes were higher in the cells overexpressing CC mutant than in those expressing wild-type Pfk1 (*Figure 5B*). We did not observe any reproducible difference in the metabolic changes between the cells overexpressing wild-type Tdh3 and its CC mutant (*Figure 5C*). We thus concluded that overexpression of Pfk1 and Tdh3 did not trigger significant metabolic changes through their catalytic activities, at least in the detected glycolytic metabolites.

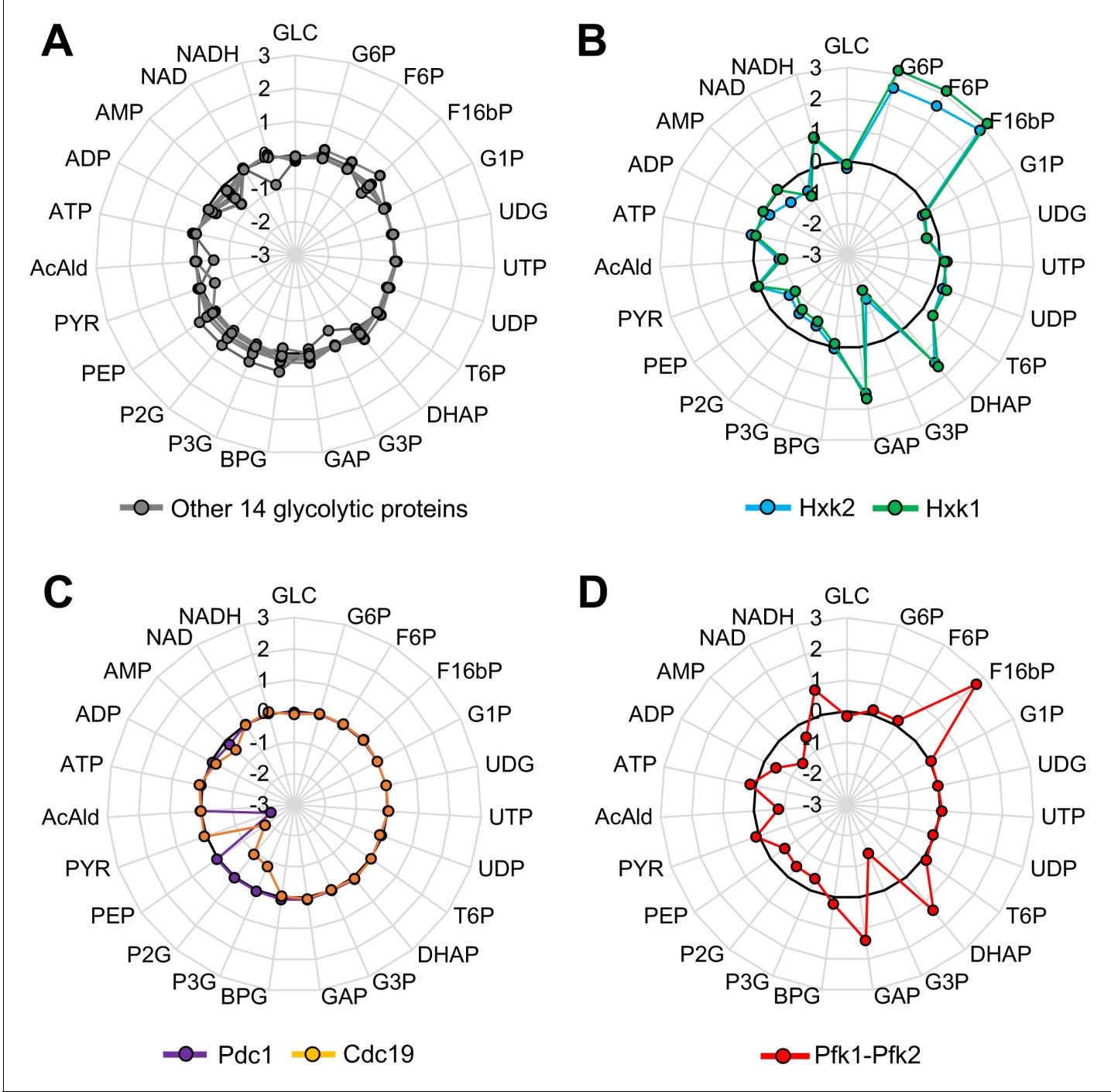

**Figure 4.** Metabolic perturbations triggered by overexpression of glycolytic proteins in silico. (A–D) Metabolic change triggered by overexpression of the indicated glycolytic protein in a kinetic model of glycolytic metabolism (*Smallbone et al., 2013*). Log$_{10}$ fold-change in each metabolite level in a simulation with 128-fold overexpression of each glycolytic protein compared to that in the wild-type is shown. In (D), Pfk1 and Pfk2 are simultaneously overexpressed.

DOI: https://doi.org/10.7554/eLife.34595.013

The following source data and figure supplement are available for figure 4:

**Source data 1.** This spreadsheet contains all data and statistical values associated with the figure.
DOI: https://doi.org/10.7554/eLife.34595.015

**Figure supplement 1.** Schematic representation of the kinetic model of glycolytic metabolism used in this study (*Smallbone et al., 2013*).
DOI: https://doi.org/10.7554/eLife.34595.014

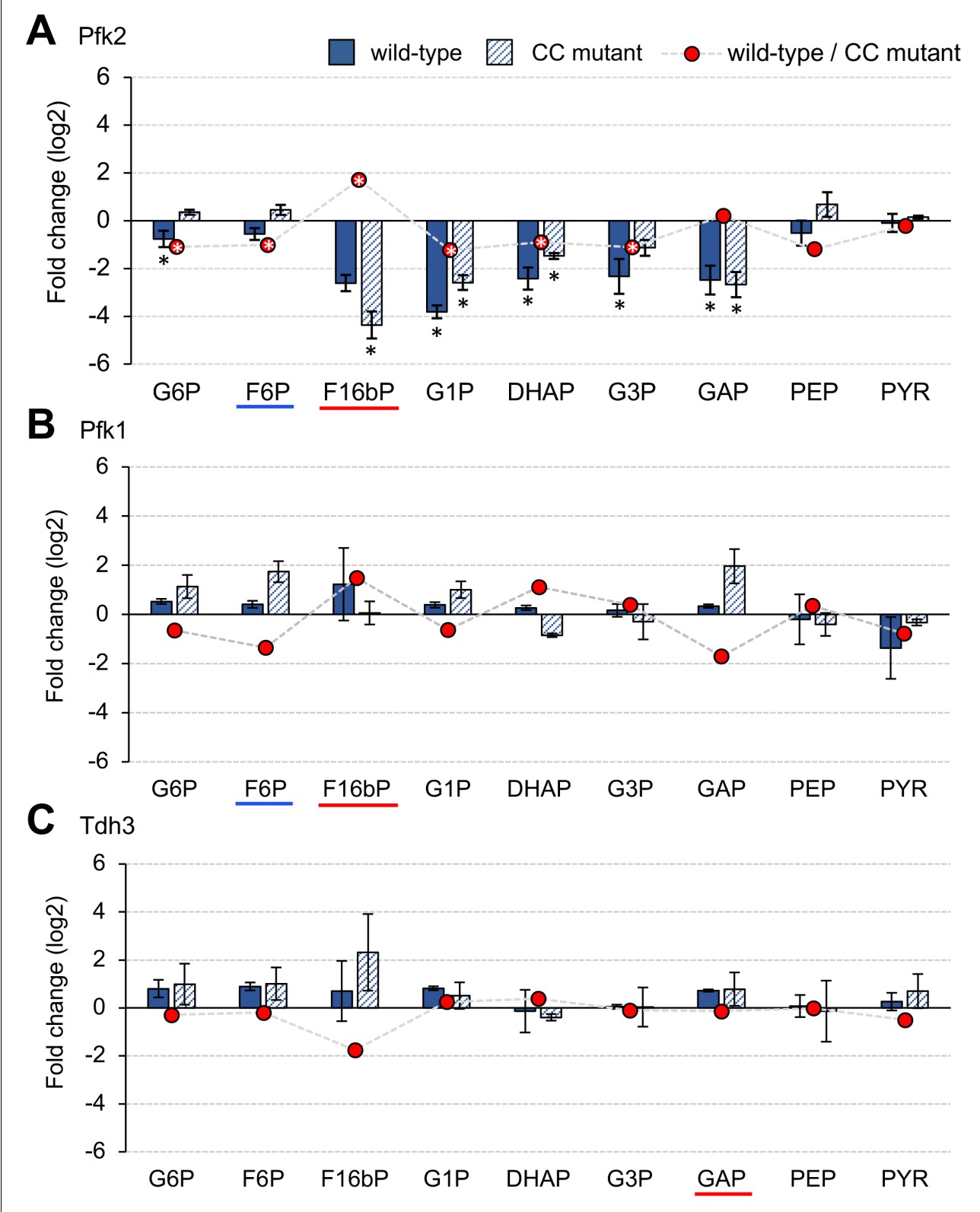

**Figure 5.** Metabolic changes triggered by overexpression of glycolytic proteins in vivo. (**A–C**) The bar graph shows the log$_2$-fold change in each metabolite in the cells overexpressing wild-type and mutant Pfk2, Pfk1, and Tdh3 over the vector controls. The red circle shows the log$_2$ fold-difference in each metabolite between the wild-type and the CC mutant measurements. The metabolites were measured in exponentially growing cells cultured in

*Figure 5 continued on next page*

Figure 5 continued

low-copy conditions. The error bars indicates the standard deviations of the mean for three (Pfk2) and two (Pfk1, Tdh3) biological replicates. *p<0.05 in Welch's *t*-test.

DOI: https://doi.org/10.7554/eLife.34595.016

The following source data and figure supplement are available for figure 5:

**Source data 1.** This spreadsheet contains all data and statistical values associated with the figure.

DOI: https://doi.org/10.7554/eLife.34595.018

**Figure supplement 1.** Comparison of metabolic changes triggered by overexpression of glycolytic proteins in silico and in vivo.

DOI: https://doi.org/10.7554/eLife.34595.017

## Codon optimality explains the lower expression of non-harmful glycolytic proteins

We next focused on Glk1, Pyk2, and Pdc6, as their expression levels were lower than those of other glycolytic proteins in low-copy conditions, while they did not seem to be harmful (*Figure 2D*). Moreover, the expression levels of Glk1 and Pyk2 were significantly elevated in high-copy conditions (*Figure 6A*). These results raised the possibility that expressed protein levels per single gene copy are lower than those for other genes either because protein synthesis rates are low or because protein degradation rates are high. Codon optimality strongly contributes totranslational elongation rate and mRNA stability (*Presnyak et al., 2015*). Therefore, we analyzed the tRNA adaptation index of a gene (tAIg) (*Tuller et al., 2010*) for the the glycolytic genes studied here (*Figure 6B* and *Figure 6—figure supplement 1*) and noticed that *GLK1*, *PYK2*, and *PDC6* had a much lower tAIg than the other glycolytic genes. To test whether the codon optimality of *GLK1* affects the protein expression level, we constructed codon-optimized *GLK1* (*CoGLK1*) and measured its protein expression level (*Figure 6A*). Glk1 expressed from *CoGLK1* was present at levels 3.6 and 4.7 times higher than that expressed from native *GLK1* in low- and high-copy conditions, respectively. We concluded that Glk1 expression was low due to its low codon optimality.

Glk1 expression increases after a diauxic shift—a growth-phase shift triggered by the carbon source alteration from glucose to ethanol (*Zampar et al., 2013*). We speculated that *GLK1* might have a codon usage that is optimized for the tRNA pool after a diauxic shift and its translational rate might be higher after the shift. To investigate this possibility, we monitored the expression levels of GFPs with different codon usages under different growth conditions. We constructed two GFP genes whose codons were differently optimized: (i) *oG-GFP*, whose codons were selected at random with probabilities obtained from the codon usage table of *GLK1*, and (ii) *oT-GFP*, whose codons were substituted by the synonymous codon used most frequently in *TDH3*. We added the ornithine decarboxylase degron (*Jungbluth et al., 2010*) to the C-terminus of these GFP genes to allow accurate monitoring of the timings of their syntheses. *Figure 6C* shows the GFP fluorescence and the growth of cells expressing the GFP genes. The GFP fluorescence of both genes peaked during their exponential growth phases. Next, we measured the lag time between the inflection points of the GFP fluorescence curve and the growth curve (where the diauxic shift is supposed to happen), as shown in *Figure 6D*. Because the lag times were not significantly different (p=0.44), we concluded that the codon usage of *GLK1* was not optimized to maximize their translation after the diauxic shift.

## Overexpression-triggered protein aggregation through S–S bonds restricts the expression limits of Tpi1

When we measured the expression levels of Eno2 and Pgk1 proteins, we unexpectedly observed high-molecular-weight bands whose sizes (~125 and 100 kDa) were different from the sizes of the monomers or dimers of Eno2 and Pgk1 (45 and 90 kDa, respectively) (*Figure 7A*). The band formation was independent of the catalytic activities of Eno2 because the bands were also observed in the experiment with Eno2 CC mutant (*Figure 7A*). The band in the Eno2 experiment seemed to be S–S-bond-connected protein aggregates because it disappeared after treatment with the reducing agent dithiothreitol (DTT) (*Figure 7B*). We confirmed that cysteines were responsible for creating these bands because they disappeared when cysteine residues were removed from Pgk1 and Eno2 (*Figure 7—figure supplement 1*). To identify the protein species in the bands, we analyzed them by liquid chromatography-tandem mass spectrometry (LC-MS/MS). As shown in *Figure 7C and D*, we

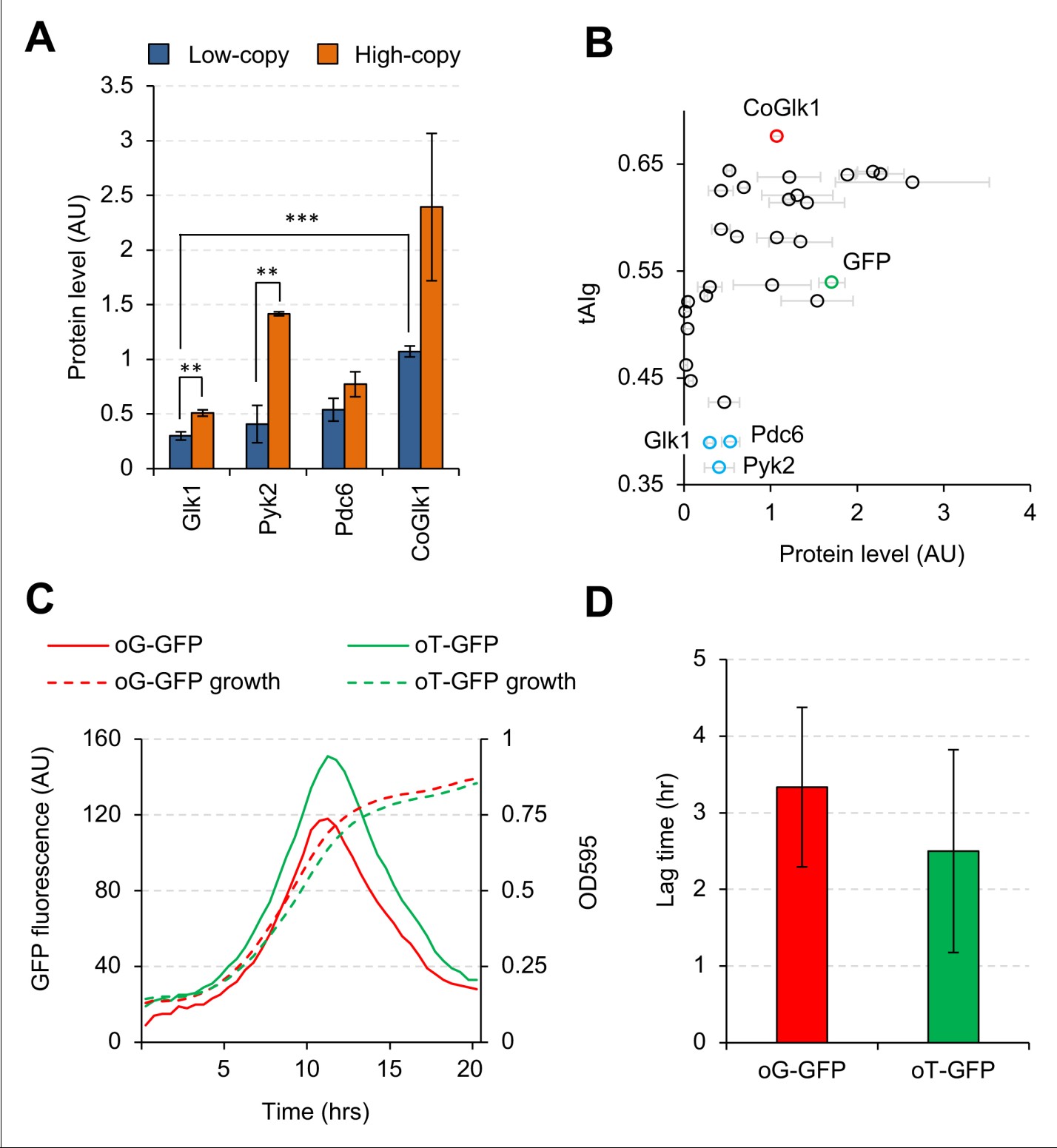

**Figure 6.** Codon usage affects the expression level, but not the synthesis timing, of Glk1. (A) Expression levels of Glk1, Pyk2, Pdc6, and codon-optimized GLK1 (CoGlk1) in the indicated conditions. (B) Relationship between the tAIg and the expression level of each glycolytic protein in low-copy conditions. Protein level data are the same as those shown in *Figure 2B*. (C) Growth curves and GFP fluorescence of cells expressing codon-optimized GFPs. *oG-GFP* (tAIg = 0.40): a GFP gene whose codons were optimized for the *GLK1* codon usage. *oT-GFP* (tAIg = 0.64): a GFP gene whose codons were optimized for the *TDH3* codon usage. (D) Lag time between the timings with the maximum GFP fluorescence and the maximum growth rate.
*Figure 6 continued on next page*

*Figure 6 continued*

Timings of the maximum GFP fluorescence and the maximum growth rate are the time points with maximum second derivatives of GFP fluorescence and growth curves. The error bars indicate the standard deviations of the means. **p<0.01; ***p<0.001 in Welch's *t*-test.

DOI: https://doi.org/10.7554/eLife.34595.019

The following source data and figure supplement are available for figure 6:

**Source data 1.** This spreadsheet contains all data and statistical values associated with the figure.

DOI: https://doi.org/10.7554/eLife.34595.021

**Figure supplement 1.** Scatter plot showing the tAlg and native expression level of *S. cerevisiae* protein.

DOI: https://doi.org/10.7554/eLife.34595.020

mainly detected glycolytic proteins, translational elongation factors, and translation initiation factors, in addition to each overexpressed protein. Most of the detected proteins were also detected in the CC mutant experiment (*Figure 7C*). This aggregation did not seem to affect the expression limits of Eno3 and Pgk1 because the expression limits of wild-type proteins and cysteine-less mutants (Eno2-C248S and Pgk1-C98S) were indistinguishable (*Figure 7—figure supplement 2*).

Next, we focused on Tpi1 because it was detected in both Pgk1 and Eno2 aggregates (*Figure 7C–D*) and because its expression limit (1.7 U) was lower than that of the highest-limit proteins such as Pgk1 and Gpm1 (>2.0 U) (*Figure 2B*). As shown in *Figure 8A*, Tpi1 constituted many aggregation bands upon its overexpression. The majority of these bands disappeared when cysteine residues were removed from Tpi1 (C41S, C126S), or after DTT treatment. These results suggested that non-specific S-S-bond-connected aggregation occurred upon overexpression of Tpi1. To test whether the aggregation restricts the Tpi1 expression limit, we measured the expression limits of cysteine-less Tpi1. As shown in *Figure 8B*, the expression levels of cysteine-less Tpi1 significantly increased above those of wild-type Tpi1. Because mutant Tpi1 levels were higher than wild-type Tpi1 levels, even in +DTT conditions, the removal of cysteine residues would not only prevent the formation of aggregates but would also increase the expression limit of Tpi1.

## Discussion

According to the protein-burden concept (*Dong et al., 1995*; *Kafri et al., 2016*; *Shah et al., 2013*; *Snoep et al., 1995*; *Stoebel et al., 2008*), the ultimate overexpression of any protein could cause growth defects by overloading basic protein production resources. But only non-harmful proteins can be overexpressed up to the ultimate level, or the protein-burden limit, because the expression limit of harmful proteins should be restricted by their harmful effects. Knowing the protein-burden limit itself is thus essential when seeking to determine whether the overexpression of a protein is harmful to cellular functions. We previously estimated the protein-burden limit of *S. cerevisiae* cells by measuring the expression level of GFP that causes growth defects. This was 15% of the total cellular protein (*Kintaka et al., 2016*).

In this study, we first tried to measure the expression limits of yeast glycolytic proteins in order to confirm whether the protein-burden limit measured using GFP applies to endogenous proteins. Most of the glycolytic proteins studied here caused growth defects when they were expressed from a strong promoter on a multicopy plasmid (*Figure 1*). The expression levels of some glycolytic proteins in these conditions were, indeed, comparative or even higher than that of GFP (*Figure 2*). Also, their expression levels did not increase due to mutations in their catalytic centers (*Figure 3A*). These results strongly suggest that the protein-burden effect largely determines the expression limit, and that the limit is around 15% of the total cellular protein. Among the glycolytic proteins studied here, Pgk1, Gpm1, and Eno2 had the highest expression limits. Although Pgk1 (44.7 kDa) and Eno2 (46.9 kDa) are 1.5-fold larger than Gpm1 (27.6 kDa), their expression limits were similar to those of Gpm1 (*Figure 2B*). These results suggest that a protein's size does not affect its expression limit, at least for proteins in this molecular weight range. These data also suggest that the expression limits of proteins are not determined by the molar concentrations of those proteins but by the cost of the protein production.

Some other glycolytic proteins, such as Pfk1, Pfk2, Adh3, and Hxts, showed expression limits far below the protein burden limit of 15% (*Figure 2*), suggesting that overexpression of these proteins is harmful. Of the 18 glycolytic proteins studied, Pfk1 and Pfk2 were the only ones whose expression

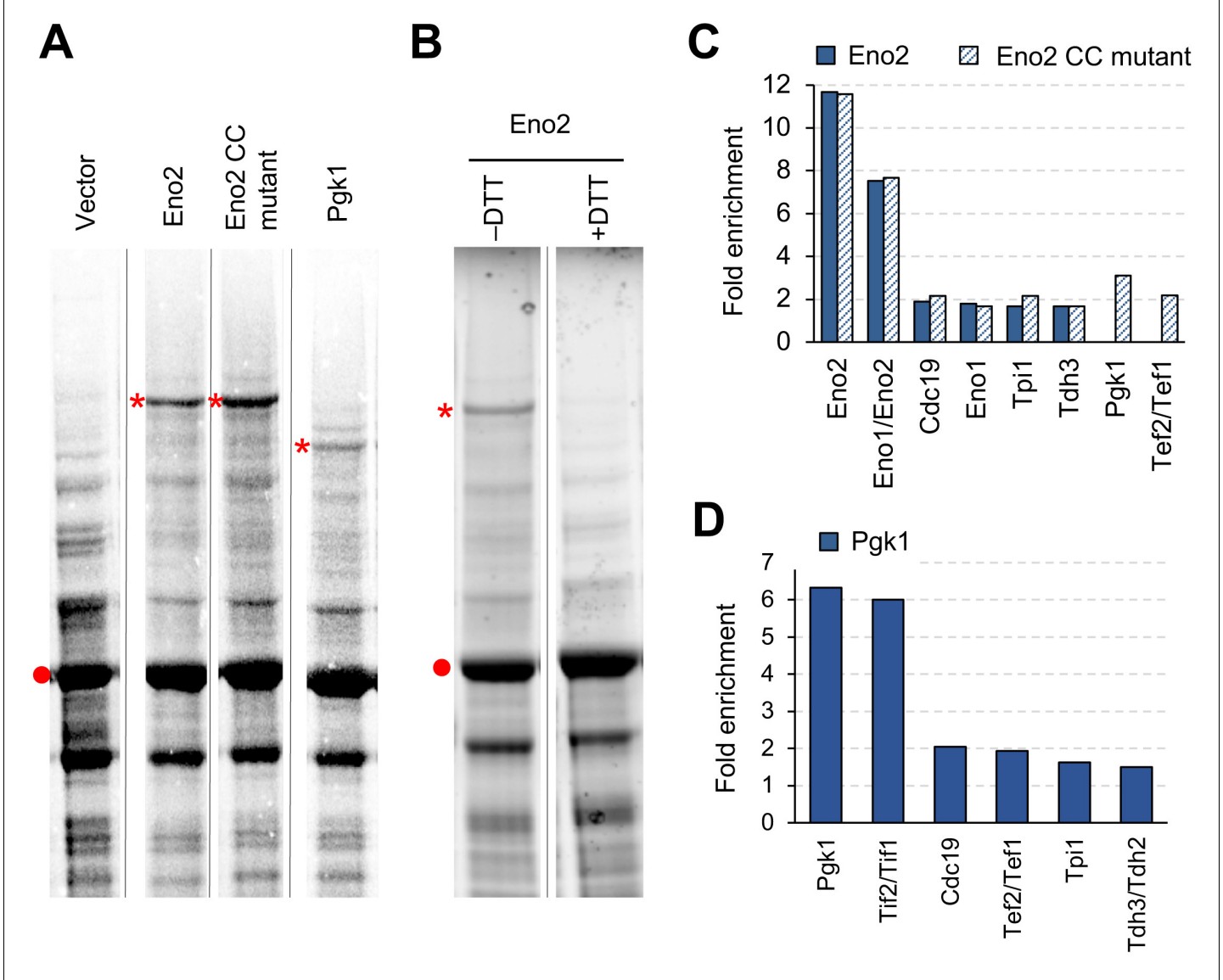

**Figure 7.** Overexpressed Eno2 and Pgk1 form protein aggregates. (A) SDS-PAGE-separated total cellular proteins from cells overexpressing the indicated proteins. (B) SDS-PAGE-separated total cellular proteins from cells overexpressing Eno2 after treatment with (+) or without (–) the reducing agent DTT. (C and D) Enriched proteins in the high molecular bands from cells overexpressing the indicated proteins. Proteins enriched in the high molecular bands > 1.5-fold over the vector control are shown. Proteins were analyzed in low-copy conditions. The red point indicates the expected molecular weight of overexpressed proteins. The asterisk indicates the high-molecular-weight band specifically observed upon overexpression of each glycolytic protein. Gel images were contrasted so that high-molecular-weight bands were visible. CC mut.: catalytic center mutant.

DOI: https://doi.org/10.7554/eLife.34595.022

The following source data and figure supplements are available for figure 7:

**Source data 1.** This spreadsheet contains all data and statistical values associated with the figure.
DOI: https://doi.org/10.7554/eLife.34595.025
**Figure supplement 1.** Aggregation of Eno2 and Pgk1 requires cysteine residues in Eno2 and Pgk1.
DOI: https://doi.org/10.7554/eLife.34595.023
**Figure supplement 2.** Effect of cysteine substitutions on the expression levels of Eno2 and Pgk1.
DOI: https://doi.org/10.7554/eLife.34595.024

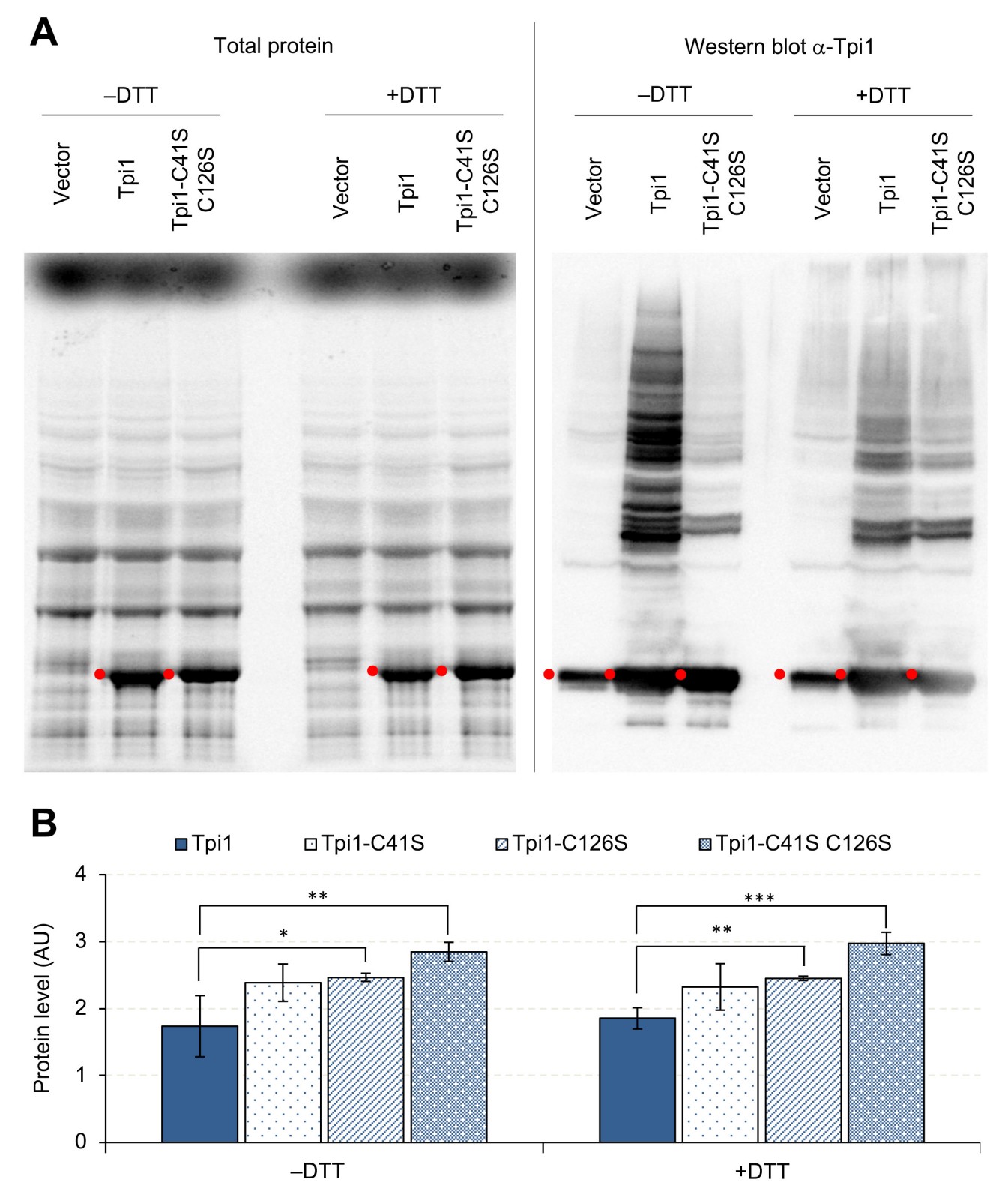

**Figure 8.** Overexpressed Tpi1 forms protein aggregates. (**A**) SDS-PAGE-separated total proteins from cells overexpressing indicated proteins, and their Western blot imaging using anti-Tpi1 antibodies. The red points indicate the expected molecular weight of overexpressed proteins. (**B**) Effect of cysteine substitutions on the expression level of Tpi1. *p<0.05; **p<0.01; ***p<0.001 in Welch's *t*-test. C41S: substitution of cysteine 41 to serine; C126S: substitution of cysteine 126 to serine. Proteins were analyzed in low-copy conditions.

*Figure 8 continued on next page*

*Figure 8 continued*

DOI: https://doi.org/10.7554/eLife.34595.026

The following source data is available for figure 8:

**Source data 1.** This spreadsheet contains all data and statistical values associated with the figure.
DOI: https://doi.org/10.7554/eLife.34595.027

limits were significantly increased by mutations in their catalytic centers (*Figure 3A–B*), suggesting that their metabolic functions restrict their expression limits. We think, however, that the metabolic perturbations that triggered the overexpression only partially affect the expression limits because the expression limits of the mutant proteins were still far below those of other glycolytic proteins (*Figure 3A–B*). Pfk1 and Pfk2 form a hetero-octameric complex, and their stoichiometric imbalance leads to the formation of filamentous Pfk1 structures in the cytosol (*Schwock et al., 2004*). This stoichiometry-imbalance-triggered protein aggregate might cause growth defects upon overexpression of Pfk1 (and Pfk2), although we could not confirm this hypothesis because simultaneous overexpression of Pfk1 and Pfk2 did not increase the expression limits of these proteins (our unpublished observation).

The CC mutants of Fba1 and Pgk1 showed lower expression limits than their wild-type proteins (*Figure 3A*). We currently do not have any substantial and consistent explanation of why these CC mutants have lower expression limits. We can assume some general mechanisms: CC mutant proteins sequester the wild-type enzymes into inactive complexes; CC mutant proteins sequester the substrate molecules for the wild-type enzymes; or mutation in the catalytic center destabilizes the structure of the enzyme. For example, Fba1 is an essential homodimeric enzyme (UniProtKB: P14540). Overexpression of CC mutant Fba1 molecules might sequester active wild-type Fba1 molecules into inactive complexes. The limit of CC mutant Tdh3 was higher than that of the wild-type in low-copy conditions whereas it was lower in high-copy conditions (*Figure 3A–B*). This strange behavior might be related to its moonlighting function. The catalytic activity of Tdh3 did not seem to explain the difference in the expression limits of wild-type and CC mutant Tdh3 (*Figure 5C*). Beside its metabolic function, Tdh3 directly binds to Sir2 protein to promote transcriptional silencing, and a mutation in the catalytic center (C150G) reduces the silencing (*Ringel et al., 2013*). It is thus possible that the CC mutant Tdh3 (C150S) causes silencing in a dose-dependent manner by competing with wild-type Tdh3 for binding with Sir2.

We speculated that the localization of Adh3 to the mitochondria and of Hxts to the plasma membrane restricted their expression limits because localized proteins overload more-limited localization resources (*Kintaka et al., 2016*). This hypothesis was confirmed because the removal of the mitochondrial signal from Adh3 increased its expression limit (*Figure 3C,D*). We also speculated that the expression limits of membrane proteins such as Hxts should be restricted by their localization, although there is no experimental evidence to support this hypothesis yet.

The fact that the expression limits of most glycolytic proteins were not affected by mutations in their catalytic centers (*Figure 3A*) suggests that their overexpression does not cause metabolic perturbations. This finding was theoretically confirmed by simulations using a kinetic model of glycolytic metabolism (*Figure 4*). The reason why their overexpression does not cause metabolic perturbations is probably that they are bidirectional enzymes: the metabolic flux should be determined only by the availability of substrates when the concentrations of these enzymes are more than a certain level. To support this idea, the overexpression of 14 bidirectional enzymes showed minor metabolic changes, whereas the overexpression of 6 unidirectional enzymes (including Hxks, Pfks, Cdc19, and Pdc1) showed strong metabolic changes in the simulation (*Figure 4*). The expression limits of Hxks in the cells, however, were close to the protein burden limit (*Figure 2B*) and were not affected by mutations in the catalytic center (*Figure 3A*). These results suggest an additional mechanism that is not implemented into the model that allows cells to avoid the effects of big metabolic changes upon overexpression of Hxks: a mechanism that prevents these metabolic perturbations from occurring, or a mechanism that prevents these metabolic perturbations from causing growth defects.

Through the metabolic analysis, we realized that we currently do not have any systematic way to identify metabolic changes that are directly triggered by the overexpression of an enzyme, because metabolism is interconnected and the overexpression of a protein could cause non-specific

perturbations that ultimately affect metabolism. Moreover, we know very little about how much change in which metabolite triggers a growth defect. Comparison of the metabolic changes in cells overexpressing wild-type and CC-mutant enzymes could be one solution for this. In fact, we observed a three-fold difference between cells expressing wild-type and CC mutant Pfk2 (*Figure 5A*). Nevertheless, once again, we cannot conclude from our current knowledge that this difference causes the difference in the expression limits of these two forms of Pfk2. By using a mathematical model, we tried to predict the potential metabolic changes that would be triggered by overexpression of an enzyme without considering unknown effects other than the enzyme's metabolic activity. In the simulations, overexpression of Pfks and Hxks triggered divergent and almost catastrophic metabolic changes (~1000-fold increase in some metabolites, *Figure 4B,D*), suggesting that their overexpression would cause growth defects due to these strong metabolic perturbations. We thus expected to obtain similar metabolic changes upon overexpression of Pfks, whose CC mutants had higher expression limits. We did not, however, observe such great changes (*Figure 5A–B* and *Figure 5—figure supplement 1*). To answer these issues precisely, we need a much deeper understanding of the connections between metabolite levels and cellular growth.

The translational rate of some glycolytic proteins, including Glk1, seemed low because of their lower codon optimality (*Figure 6*). Actually, the codon optimality of Glk1 (tAIg = 0.38) is close to the average for all the yeast genes (tAIg = 0.37), and the codon optimality of other glycolytic proteins studied here is exceptionally high (*Figure 6—figure supplement 1*). These observations suggest that the codon optimality of most yeast genes is not high enough to allow expression of their proteins up to the protein-burden limit, even if they are expressed from a strong promoter on a multi-copy plasmid.

Overexpression of Eno2, Pgk1, and Tpi1 triggered S–S-bond-connected aggregation (*Figures 7* and *8*), and the aggregates that are formed contain other glycolytic proteins and translational factors (*Figure 7C–D*). We think that this aggregation is triggered by spontaneous non-specific S–S bond formation among proteins existing in high concentrations. Interestingly, we also detected the same proteins within the gel of the corresponding molecular weight in the vector control, although the amounts estimated by LC-MS/MS were lower and cannot be identified as visible protein bands (*Figure 7—source data 1*). Therefore, we speculated that the S–S-bond-mediated protein aggregation occurs even in normal physiological conditions, but it is accelerated by an increase in the concentration of cytoplasmic proteins upon overexpression of glycolytic proteins. This aggregation might affect the expression limits of cysteine-containing glycolytic proteins, because changing the cysteine residues of Tpi1 into serine residues increases the protein's expression limit (*Figure 8B*). As the amount of protein corresponding to the Tpi1 monomer was not changed by DTT treatment, the expression level of Tpi1 should not be reduced simply by aggregation but by the harmful effect of spontaneous S–S bond formation. This hypothesis is supported by the fact that the most highly expressed glycolytic protein Gpm1, which has a molecular weight similar to that of Tpi1, does not have a cysteine residue. The deleterious effect of this aggregation, however, seems protein-specific because the expression limits of Pgk1 and Eno1 were among highest measured (*Figure 2A*), and removal of their cysteine did not increase their expression limits (*Figure 7—figure supplement 1*).

As described above, we revealed mechanisms that restrict the expression limits of some glycolytic proteins. We do not think, however, that these mechanisms are the sole factors restricting the expression limits of these proteins. The expression limits of ΔMTS-Adh3 (0.45 AU, *Figure 3D*) and CoGlkl (1.07 AU, *Figure 6A*) are still lower than those of other high limit proteins such as Pgk1 and Gpm1 (2.26 AU and 2.63 AU, respectively, *Figure 2B*). It is thus likely that multiple mechanisms restrict the expression limits of these proteins.

Protein misfolding or misinteraction is considered to cause toxicity upon high-level expression of a protein with low translational robustness, low folding stability, or a high propensity for misinteraction (*Drummond and Wilke, 2009*; *Zhang and Yang, 2015*). In general, highly expressed proteins such as glycolytic proteins are thus evolved to avoid these characteristics (*Zhang and Yang, 2015*), and that should be a requirement for a protein to be expressed up to the protein-burden limit. Cdc19, one of the glycolytic proteins studied here, aggregates in a stress-induced and reversible manner through a region of low compositional complexity (*Saad et al., 2017*). This aggregation capacity of Cdc19 might explain why its expression limit (0.42 AU) is lower than the protein burden limit (>2.0 AU) (*Figure 2B*). Our finding in *Figure 8* suggested that the high-level expression of a cysteine-containing protein could also cause a misinteraction-triggered toxic effect; hence

unimportant cysteines should be avoided in highly expressed proteins. Concentration-dependent liquid phase separation is also considered to cause toxicity upon overexpression of structurally disordered and nucleic-acid-binding proteins (*Bolognesi et al., 2016*). We do not think that this mechanism caused growth defects upon overexpression of the glycolytic proteins studied here because they are less structurally disordered (*Moriya, 2015*) and not nucleic-acid-binding proteins.

We summarize our analysis in *Supplementary file 1*. In conclusion, we established the ultimate expression level that causes cellular growth defects due to the protein-burden effect as around 15% of the total cellular protein. The next interesting theme is to identify characteristics of proteins that can be overexpressed up to the protein-burden limit because such proteins are considered non-harmful to cellular functions. Those characteristics should conversely imply the properties of proteins that are harmful when they are overexpressed.

## Materials and methods

### Strains, growth conditions, and yeast transformation

BY4741 (*MATa his3Δ1 leu2Δ0 met15Δ0 ura3Δ0*) (*Brachmann et al., 1998*) was used as the host strain for the experiments. Yeast culture and transformation were performed as previously described (*Amberg et al., 2005*). A synthetic complete (SC) medium without uracil (Ura) or leucine (Leu), as indicated, was used for yeast culture.

### Plasmids used in the study

The plasmids used in the study are listed in the Key Resources Table (*Supplementary file 2*). The plasmids were constructed by the homologous recombination activity of yeast cells (*Oldenburg et al., 1997*), and their sequences were verified by DNA sequencing.

### Measurement of the plasmid copy number

The plasmid copy number was measured by real-time polymerase chain reaction, as previously described (*Moriya et al., 2006*), using a LightCycler480 system (Roche). The LEU2 (LEU2-2F and LEU2-2R) and LEU3 primer sets (LEU3-3F and LEU3-3R) were used to amplify DNA fragments of the pTOW40836 plasmid and genomic DNAs, respectively. Mean values, standard deviations (SD), and *p*-values of Welch's *t*-test were calculated from biological triplicates.

### Protein analysis

The total protein was extracted from log-phase cells with an NuPAGE LDS sample buffer (Thermo-Fisher) after 0.2N NaOH treatment (*Kushnirov, 2000*). For each analysis, the total protein extracted from two optical density (OD) units of cells with $OD_{600}$ was used. For total protein visualization, the extracted total protein was labeled with Ezlabel FluoroNeo (ATTO), as described in the manufacturer's protocol, and separated by 4–12% SDS-PAGE. Proteins were detected and measured using the LAS-4000 image analyzer (GE Healthcare) in SYBR–green fluorescence detection mode and Image Quant TL software (GE Healthcare). The expression of each target protein (AU) was calculated, as shown in *Figure 2*. Average values, SD, and *p*-values of Welch's *t*-test were calculated from biological triplicates. For detection of Tpi1, the SDS-PAGE-separated proteins were transferred to a PVDF membrane (ThermoFisher). Tpi1 was detected using an anti-Tpi1 antibody (RRID:AB_11130951), a peroxidase-conjugated secondary antibody (Nichirei Biosciences), and a chemiluminescent reagent (ThermoFisher). The chemiluminescent image was acquired with an LAS-4000 image analyzer in chemiluminescence detection mode.

### Measuring growth rate and GFP fluorescence

Cellular growth and GFP fluorescence were measured by monitoring $OD_{595}$ and Ex485 nm/Em 535 nm, respectively, every 30 min using an Infinite F200 microplate reader (Tecan). The maximum growth rate (MGR) was calculated as described previously (*Moriya et al., 2006*). Average values, SD, and *p*-values of Welch's *t*-test were calculated from biological triplicates. We define growth defect based on a significant reduction in the maximum growth rate of the cells overexpressing a target protein compared with that of cells overexpressing the control vector (p<0.01, Welch's t-test).

### In silico analysis of overexpression of glycolytic proteins

We used a kinetic model of the yeast glycolytic pathway developed previously (*Smallbone et al., 2013*). To predict metabolic changes upon overexpression of glycolytic proteins, we changed the initial concentration of each target protein 128-fold over the original concentration, and calculated the concentration of each metabolite at the steady state. We did not analyze the metabolism for the overproduction of Pyk2, Adh2, Adh3, Adh4, and Adh5, because they were not included or because their turnover ratios were set to 0 in the model. We also did not analyze Hxts overexpression, because its concentration was not changeable in the model.

### Metabolite analysis

Yeast cells were aerobically cultivated at 30°C for 24–48 hr in an SC–Ura medium. The cells were inoculated into 200 mL of the medium at an $OD_{600}$ of 0.5 and then aerobically cultured at 30°C for 3 hr. 1.0 mL of culture containing cells with an of $OD_{600}$ of 50 was mixed with 1.4 mL of methanol solution pre-cooled at –80°C. The sample was centrifuged at 5,000 g at –20°C for 5 min. After the removal of the supernatant, 1.0 mL of 75% ethanol pre-heated at 95°C was added to the sample, which was then incubated for 3 min at 95°C. 10 μL of 17 μM D-camphor sulfonic acid was added to the sample as an internal standard for liquid chromatography triple–stage quadrupole-mass spectrometry (LC-QqQ-MS) analysis. After placing on ice for 5 min, the sample was centrifuged at 5,000 g at 4°C for 5 min to remove cell debris. 950 μL of the supernatant was transferred to a new tube and centrifuged at 15,000 rpm at 4°C for 5 min. 300 μL of the supernatant collected as cell extract was dried under vacuum, and then stored at –80°C until the mass spectrometry analysis. All metabolites were measured using LC-QqQ-MS. LC-QqQ-MS analysis was performed according to the method given by *Kato et al. (2012)*. We calculated the normalized internal standard peak areas for each metabolite. Samples from three independent cultures were analyzed for the cells overexpressing Pfk2, Pfk2 CC mutant, and the vector control. Samples from two independent cultures were analyzed for the cells overexpressing Pfk1, Pfk1 CC mutant, Tdh3, Tdh3 CC mutant, and the vector control.

### Identification of aggregated protein species

The total protein extracts in the overexpression of Eno2, Eno2 CC mutant, and Pgk1 were separated by SDS-PAGE and stained by Coomassie staining solution (ThermoFisher). Proteins of interest were excised from the gels and digested using trypsin. The tryptic peptides were analyzed by LC-MS/MS consisting of an LTQ-Orbitrap mass spectrometer (ThermoFisher) and a DiNa nano LC (KYA Technologies) system according to the method described previously (*Kito et al., 2016*). The peptide mixture was separated with reverse-phase chromatography. Mobile phase A contained 0.1% formic acid, and mobile phase B contained 0.1% formic acid/80% acetonitrile. Peptides were eluted at a flow rate of 200 nL/minute using a 55 min gradient as follows: from 0% to 32% solvent B over 45 min, from 32% to 40% solvent B over 5 min, and from 40% to 80% solvent B over 5 min. The acquired MS/MS spectra were subjected to a database search against the protein sequences of *S. cerevisiae*. The aggregating protein species in *Figure 7* are those for which the number of peptide hits in the database search was five or more and was 1.5-fold more than that of the vector control.

## Acknowledgements

We thank the members of the Moriya laboratories for advice and helpful discussions, Mr. Katsuhiro Yamamoto and Ms. Yoshimi Hori for experimental support, and Dr. Kei Takahashi for providing experimental materials.

# Additional information

## Funding

| Funder | Grant reference number | Author |
| --- | --- | --- |
| New Energy and Industrial Technology Development Organization | Development of Production Techniques for Highly Functional Biomaterials Using Smart Cell, P16009 | Hisao Moriya |
| Japan Society for the Promotion of Science | KAKENHI 17H03618 | Hisao Moriya |
| Japan Society for the Promotion of Science | KAKENHI 15KK0258 | Hisao Moriya |

The funders had no role in study design, data collection and interpretation, or the decision to submit the work for publication.

## Author contributions

Yuichi Eguchi, Data curation, Investigation, Writing—original draft, Writing—review and editing; Koji Makanae, Data curation, Writing—review and editing; Tomohisa Hasunuma, Keiji Kito, Data curation, Validation, Methodology, Writing—review and editing; Yuko Ishibashi, Data curation, Methodology; Hisao Moriya, Conceptualization, Data curation, Supervision, Funding acquisition, Investigation, Writing—original draft, Writing—review and editing

## Author ORCIDs

Yuichi Eguchi (iD) http://orcid.org/0000-0002-4809-1402
Hisao Moriya (iD) http://orcid.org/0000-0001-7638-3640

## Decision letter and Author response

Decision letter https://doi.org/10.7554/eLife.34595.032
Author response https://doi.org/10.7554/eLife.34595.033

# Additional files

## Supplementary files

• Supplementary file 1. Summary table showing analyzed glycolytic proteins, their characteristics, analysis types, quantitative interpretation of the results, and proposed mechanisms to restrict expression limits of glycolytic proteins.
DOI: https://doi.org/10.7554/eLife.34595.028

• Supplementary file 2. Key resources table.
DOI: https://doi.org/10.7554/eLife.34595.029

• Transparent reporting form
DOI: https://doi.org/10.7554/eLife.34595.030

## Data availability

All data generated or analysed during this study are included in the manuscript and supporting files.

The following datasets were generated:

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
