## [Decision Letter]

Thank you for submitting your article "Estimating the Protein Burden Limit of Yeast Cells by Measuring Expression Limits of Glycolytic Proteins" for consideration by *eLife*. Your article has been reviewed by three peer reviewers, and the evaluation has been overseen by a Reviewing Editor and Detlef Weigel as the Senior Editor. The following individual involved in review of your submission has agreed to reveal his identity: Claus O Wilke (Reviewer #3).

The reviewers have discussed the reviews with one another and the Reviewing Editor has drafted this decision to help you prepare a revised submission.

Summary:

Moriya et al., attempt to determine the protein expression limits of glycolytic proteins. They measure the growth rate, protein expression and copy number of glycolytic proteins using both high copy and low copy plasmids. They go on to estimate the contributions of various factors that might determine the upper limit of protein expression such as metabolic activity, codon usage, membrane localization and disulfide bond formation. The authors conclude that while metabolic activity has no role in determining the expression limit, disulfide bond formation, membrane localization and sub-optimal codon usage limit the extent of protein expression. The authors assert the ability of their pipeline to differentiate between proteins that can be overexpressed from those that will be harmful upon overexpression.

Essential revisions:

1) One of the main findings of the paper is that protein expression limits are not determined by the metabolic activity. The authors base this primarily on the lack of a difference between expression levels of proteins and their catalytic mutants (proteins with mutations in their catalytic sites). The authors need to verify that metabolism is indeed altered to unequivocally comment on the link between metabolic activity and protein expression levels. Instances that highlight the need to perform metabolic measurements in the manuscript are:

A) The authors measured metabolic activity for only one protein-mutant pair (Pfk2, Figure 4), which did not show any difference in protein expression levels (Figure 3A). The authors need to measure the metabolic levels for pairs of proteins that show a significant difference in their expression levels like Pfk1 or Tdh3 to be able to comment on the link between metabolism and protein levels.

B) In addition, there are several contradictions in the effect of these catalytic mutants on protein expression levels. Figure 3A shows that in two cases the mutant has higher expression (Pfk1 and Tdh3) but in other two cases, the wild-type has higher expression (Fba1, Eno1). How do the authors explain such differences? In addition, Tdh3 mutant has lower expression level compared to wild-type when expressed from a high copy plasmid. How do the authors explain this flip?

C) Discussion, fifth paragraph: The authors claim that one of the reasons why they don't see any association between metabolic activity and expression is that the majority of these enzymes are bidirectional. This is not true for all the enzymes as some glycolytic enzymes are unidirectional. In addition, the authors need to show a control example where unidirectional enzyme has a higher protein expression in order to make any claim between enzymatic directionality and protein expression.

2) The authors show that three mechanisms namely codon optimization, membrane localization and disulfide bond formation, determine the limit of expressions of several proteins. However, the differences observed by the authors are significant but really small. While it is likely that multiple mechanisms would contribute to determining the upper limit of protein expression, the authors need to be cautious about claiming them as the sole factors limiting expression levels (subsection “Mitochondrial localization restricting the expression limit of Adh3” and subsection “Lower expression of nonharmful glycolytic proteins explained by their codon optimality”, first paragraph). In addition, the authors claim that disulfide bonds limit the expression of Eno2 and Pgk1 by triggering aggregation. They show that addition of DTT removes the bond formation in Eno2, and changing cysteine to serine in a third protein (Tpi1) reduces its expression levels. Both these pieces of evidence are incomplete independently. Instead of performing the estimations in two different proteins, the authors need to alter cysteine to serine in Eno2 and Pgk1 and then show that the bands disappear in addition to increase in expression. Alternately, the authors need to show that Tpi1 also forms bonds which disappear upon treatment with DTT. The fact that expression of Tpi1 is independent of DTT contradicts that role of disulfide bonds in limiting expression.

3) Overall the authors play a bit fast and loose with their statistics. First, p values should be accurately reported. Don't write "p<0.05", write "p=0.032". Second, whenever p values are stated it should also be stated what test was used (see e.g. subsection “Mutations in catalytic centers not affecting expression limits of most glycolytic proteins”). Third, correlations should be reported with p values (e.g. subsection “Metabolic perturbations triggered upon overexpression of glycolytic proteins”, last paragraph).

4) While the authors describe an interesting system to estimate the limits of protein expression within a cell, there are several discrepancies between vector copy number and measured expression levels, which raises the concern the results can be reflective of the technical experimental setting instead of true limitations. In addition, the authors use GFP, an exogenous protein with a high expression, as control. Endogenous proteins, preferably unidirectional and bidirectional enzymes, that show high and low expression levels, will make for better controls to the glycolytic enzymes. The following are specific examples of such discrepancies:

A) In Figure 1B and C, why is the maximum growth rate between high and low copy number vector control so different?

B) Subsection “Measurement of expression limits of glycolytic proteins”, second paragraph: The two explanations mentioned by the authors are not mutually exclusive. The authors argue that proteins with low expression and low copy number are harmful for the cell and the ones with low expression and high copy number are repressed due to their high copy number. This is a circular argument and doesn't explain why the copy numbers are high in the first place. Finally, is the Pearson correlation of 0.3 significant? The authors have dismissed it but they need to show that it is statistically not significant.

C) The authors claim that 15 percent of the total cellular protein is the limit for overexpression of protein. However, the authors do not observe any correlation between molecular weight and protein expression levels. How do the authors explain this lack of correlation?

5) The results depend on how exactly one defines "growth defect" and how accurately one measures it. The paper does not discuss this issue. "growth defect" needs to be defined precisely, and the authors also need to argue that they can measure it with sufficient accuracy. One way by which one could get the result that there are no growth defects even at high levels of overexpression is by using a very insensitive assay.

6) A simple summary table presenting the expression limit and proposed mechanism of toxicity (or not) and the evidence for this for all 29 proteins would be very helpful. This could replace some of the information in Table 1 which could be moved to the supplement.

7) In places it is not entirely clear why the authors are only performing mechanistic experiments on a specific subset of the proteins. Again a summary table might help to better communicate what has been tested for which proteins and why.

8) 'Repression' implies an active mechanism to lower protein concentration whereas it is just that these proteins are not using optimised codons that increase translation like the other enzymes. It is better to avoid this word and simply talk about 'lower expression'.

9) The metabolic profiling is rather inconclusive. Is the conclusion simply that changes in the quantified metabolites cannot be causing the growth defect? There also doesn't seem to be much of a connection between the results of the computational simulations and the metabolic profiling, so it's not at all clear how useful the simulations are.

10) There is an obvious other source of potential growth defects that have been discussed widely in the literature but that aren't mentioned at all: Toxic effects due to protein misfolding or misinteractions. For example, Geiler-Samerotte et al. measured the effect of overexpressed, misfolded GFP on yeast growth and found an effect in proportion to the amount of misfolded protein (https://doi.org/10.1073/pnas.1017570108). Also concentration-dependent liquid demixing e.g. Bolognesi et al., 2016. Similar topics have been discussed in the literature for a long time, see e.g. this review: https://www.nature.com/articles/nrg2662. No additional work required, but discussion is needed.

---

## [Author Response]

Essential revisions:1) One of the main findings of the paper is that protein expression limits are not determined by the metabolic activity. The authors base this primarily on the lack of a difference between expression levels of proteins and their catalytic mutants (proteins with mutations in their catalytic sites). The authors need to verify that metabolism is indeed altered to unequivocally comment on the link between metabolic activity and protein expression levels. Instances that highlight the need to perform metabolic measurements in the manuscript are:A) The authors measured metabolic activity for only one protein-mutant pair (Pfk2, Figure 4), which did not show any difference in protein expression levels (Figure 3A). The authors need to measure the metabolic levels for pairs of proteins that show a significant difference in their expression levels like Pfk1 or Tdh3 to be able to comment on the link between metabolism and protein levels.

We agree the reviewer’s comment that we need a positive example of which metabolic perturbation mainly determines its expression limit. Overexpression of that protein should cause strong growth defects, and thus expression limit is low, and its CC mutation dramatically increases its expression limit up to the level of 15% of total protein (the protein burden limit). We expected to obtain this type of protein during our analysis of glycolytic proteins because such enzyme had never been identified as far as we know. However, we could not obtain it. We need to further survey whether our finding, “catalytic activity does not determine the expression limit of a metabolic enzyme”, can generally be applicable for other metabolic enzymes.

From the comparison of expression limits between wild-type proteins and catalytic center (CC) mutant proteins (Figure 3A and 3B), we currently believe that metabolic perturbations triggered by the overexpression of Pfk1 and Pfk2 (but not Tdh3) partially restrict their expression limits. We agree that difference in the expression limits between the wild-type and CC mutant is not a direct way to show whether the metabolic perturbation is a significant determinant of the expression limit, and measurement of metabolic levels would strengthen our arguments.

We first carefully re-examined our metabolite measurements of the cells overexpressing wild-type and CC mutant of Pfk2.The levels of some glycolytic metabolites were significantly lower than the those of the vector control (new Figure 5A, *p* < 0.05, Welch’s *t*-test). We further found that the level of F16bP in the cells overexpressing wild-type Pfk2 was >3-fold higher than CC mutant Pfk2 (*p* < 0.05, Welch’s *t*-test). Because F16bP is the product of Pfk catalysis and the simulation predicted the dramatic increase in the F16bP level upon overexpression of Pfks, the catalytic activity of Pfk2 might trigger this metabolic difference. However, as discussed later, it is not easy to conclude whether this minor F16bP change causes growth defects.

We then measured metabolic changes in the cells overexpressing Pfk1 and Tdh3 (and their CC mutants) in low-copy conditions where CC mutants showed increased expression limits. As shown in the new Figure 5B, C, levels of glycolytic metabolites in the cells overexpressing wild-type Pfk1 and Tdh3 were not changed more than threefold over the vector control. We did not observe reproducible higher F16bP level in the cells overexpressing wild-type Pfk1 than CC mutant Pfk1. Moreover, the metabolic changes were higher in the cells overexpressing CC mutant than wild-type Pfk1 (Figure 5B). We did not observe any reproducible difference between the metabolic changes between the cells overexpressing wild-type and those overexpressing CC mutant Tdh3 (Figure 5C). We thus concluded that overexpression of Pfk1 and Tdh3 did not trigger significant metabolic changes through their catalytic activities at least in the detected glycolytic metabolites.

We added these results in new Figure 5 and Figure 5—figure supplement 1 and thoroughly rewrote relevant descriptions in Results as follows:

“To further characterize physiological conditions triggered by the overexpression of Pfks, we next analyzed metabolic changes in yeast cells overexpressing wild-type and CC mutant Pfk2 over the vector control by measuring 35 metabolites (Figure 5–source data 1), because the CC mutants showed increased expression limits (Figure 3B). […] We thus concluded that overexpression of Pfk1 and Tdh3 did not trigger significant metabolic changes through their catalytic activities at least in the detected glycolytic metabolites.”

B) In addition, there are several contradictions in the effect of these catalytic mutants on protein expression levels. Figure 3A shows that in two cases the mutant has higher expression (Pfk1 and Tdh3) but in other two cases, the wild-type has higher expression (Fba1, Eno1). How do the authors explain such differences? In addition, Tdh3 mutant has lower expression level compared to wild-type when expressed from a high copy plasmid. How do the authors explain this flip?

Higher expression limits in CC mutants could be explained as they cause less perturbation in the metabolism, which is the reason why we constitute the mutants as described.

During the revision process, we found a mistake in the construction of the CC mutant of *PGK1* (we had introduced a synonymous mutation (115A>C) which did not change the target amino acid). We thus re-constituted a correct CC mutant of PGK1 (R39A, 115A>G, 116G>C). We are very sorry for this mistake (we checked and confirmed that we constructed all others as we had intended). We measured the expression limit of the mutant and noticed it also showed significantly lower expression limit than the wild-type (Figure 3A). We also performed statistical analysis again by more strictly (Welch’s *t*-test instead of Student’s *t*-test), and noticed the difference between the wild-type and the CC mutant of Eno1 was not significant. The CC mutants of Fba1 and Pgk1 thus showed significant lower expression limits than the wild-types.

We currently do not have any substantial and consistent explanation why these CC mutants have lower expression limits. We can assume some general mechanisms; CC mutant proteins sequester the wild-type enzymes into inactive complexes; CC mutant proteins sequester substrate molecules for the wild-type enzymes; mutation in the catalytic center destabilize the structure of the enzyme. For example, Fba1 is an essential homodimeric enzyme (UniProtKB: P14540). Overexpressed of Fba1 CC mutant molecules might sequester active wild-type Fba1 molecule into inactive complexes.

The strange behavior of Tdh3 CC mutant might be related to its moonlighting function. As described above, the difference in the expression limits between wild-type and CC mutant Tdh3 seemed not to be explained its catalytic activity. Beside its metabolic function, Tdh3 directly bind to Sir2 protein to promote transcriptional silencing (Ringel AE., et al., 2013). A mutation in the catalytic center (C150G) reduces the silencing. It is thus possible that our CC mutant of Tdh3 (C150S) affect the silencing in a dose-dependent manner by competing with the wild-type Tdh3 for binding with Sir2.

We added these discussion in Discussion section as follows:

“The CC mutants of Fba1 and Pgk1 showed lower expression limits than their wild-types (Figure 3A). […] It is thus possible that the CC mutant Tdh3 (C150S) affect the silencing in a dose-dependent manner by competing with the wild-type Tdh3 for binding with Sir2.”

C) Discussion, fifth paragraph: The authors claim that one of the reasons why they don't see any association between metabolic activity and expression is that the majority of these enzymes are bidirectional. This is not true for all the enzymes as some glycolytic enzymes are unidirectional. In addition, the authors need to show a control example where unidirectional enzyme has a higher protein expression in order to make any claim between enzymatic directionality and protein expression.

Our claim is opposite. We are sorry for our confusing descriptions. We tried to claim that overexpression of a bidirectional enzyme does not strongly affect the metabolite levels because the substrate/product levels by themselves should determine the flux. Conversely, overexpression of a unidirectional enzyme should strongly affect the metabolite levels because the enzymatic activity should determine the flux. We thus think that unidirectional enzymes (Hxks, Pfks, and Cdc19) should instead have lower expression limits. The idea is based on the results of the simulation shown in Figure 4. In the simulation, overexpression of 14 bidirectional enzymes showed minor metabolic changes (Figure 4A), while all six unidirectional enzymes showed strong metabolic changes (Figures 4B–D). At least in theoretical level, above hypothesis that overexpression of a bidirectional enzyme does not strongly affect the metabolite levels seemed right. This hypothesis, however, was not supported by the experiment because both wild-type and CC mutant of Hxts had high limits and CC mutant Cdc19 did not increase its expression limits (Figure 3A). We need further analysis about this issue, but it is beyond the scope of this study.

We thus added descriptions about this issue in Discussion:

“To support this idea, overexpression of 14 bidirectional enzymes showed minor metabolic changes, while overexpression of 6 unidirectional enzymes (Hxks, Pfks, Cdc19, and Pdc1) showed strong metabolic changes in the simulation (Figure 4). […] These results suggest an additional mechanism that is not implemented into the model to avoid big metabolic changes upon overexpression of Hxks; a mechanism that prevents these metabolic perturbations from occurring, or a mechanism that prevents these metabolic perturbations from causing growth defects.”

2) The authors show that three mechanisms namely codon optimization, membrane localization and disulfide bond formation, determine the limit of expressions of several proteins. However, the differences observed by the authors are significant but really small. While it is likely that multiple mechanisms would contribute to determining the upper limit of protein expression, the authors need to be cautious about claiming them as the sole factors limiting expression levels (subsection “Mitochondrial localization restricting the expression limit of Adh3” and subsection “Lower expression of nonharmful glycolytic proteins explained by their codon optimality”, first paragraph).

We agree that multiple mechanisms would contribute to determining expression limit of a protein, and in this study, we tried to reveal them one by one using glycolytic proteins as model proteins. We are sorry that our descriptions gave an impression that we are claiming our findings as the sole factors limiting expression levels.

We thus added some arguments about this in Discussion section as follows:

“As described above, we revealed mechanisms restricting the expression limits of some glycolytic proteins. We, however, do not think that these mechanisms are the sole factors restricting expression limits of these proteins. The expression limits of ΔMTS-Adh3 (0.45 AU, Figure 3D) and CoGlkl (1.07 AU, Figure 6A) are still lower than the other high limit proteins such as Pgk1 and Gpm1 (2.26 AU and 2.63 AU, Figure 2B). It is thus likely that multiple mechanisms would restrict the expression limits of these proteins.”

In addition, the authors claim that disulfide bonds limit the expression of Eno2 and Pgk1 by triggering aggregation. They show that addition of DTT removes the bond formation in Eno2, and changing cysteine to serine in a third protein (Tpi1) reduces its expression levels. Both these pieces of evidence are incomplete independently. Instead of performing the estimations in two different proteins, the authors need to alter cysteine to serine in Eno2 and Pgk1 and then show that the bands disappear in addition to increase in expression. Alternately, the authors need to show that Tpi1 also forms bonds which disappear upon treatment with DTT. The fact that expression of Tpi1 is independent of DTT contradicts that role of disulfide bonds in limiting expression.

As described above, we found a mistake in the construction of the CC mutant of Pgk1. We thus withdrew the identification of proteins within the aggregate observed when Pgk1 CC mutant was overexpressed from Figure 6D (now Figure 7D) as it was not the mutant protein. We are sorry for our mistake, but our finding is not primarily affected by this.

We constructed mutants of Eno2 (C248S) and Pgk1 (C98S) whose cysteine was changed into serine and analyzed their effects. We confirmed that the aggregation bands were disappeared as shown in Figure 7—figure supplement 1. We, however, did not observe any increase in the expression limits of Eno2 and Pgk1 even when their cysteines were removed. We thus concluded that S-S bond-connected proteins aggregation does not restrict the expression limits of Eno2 and Pgk1. We are not surprised by this result because only small part of overexpressed Eno2 and Pgk1 constitute the aggregation bands (the bands are only observed with a long-exposure), and expression limits of wild-type Eno2 and Pgk1 are the highest levels among other glycolytic proteins (Figure 2B); hence their expression limits seems not restricted by other mechanisms than the protein burden effect. We thus concluded that deleterious effect of this aggregation might be protein-specific.

Detection of the S-S bond connected protein aggregation was just a hint to think the reason why the expression limits of Tpi1 is far lower than Gpm1 who does not contain any cysteine (Figure 2B). Tpi1 was detected in both Eno2 and Pgk1 overexpression-triggered aggregates although Tpi1 itself was not overexpressed in that situation, suggesting that Tpi1 is a naturally-aggregative protein. As the reviewer has suggested, we tried to detect aggregation band upon overexpression of Tpi1, and if the band is disappeared when cysteines are substituted.

We did not identify any aggregation band upon overexpression of Tpi1 by the total protein staining (new Figure 8A). We want to emphasize that our critical finding in our identification of the proteins in the aggregates is that the aggregation bands contain many different proteins in addition to the overexpressed proteins themselves. Hence, the aggregation bands visible in the gel are constructed by chance, and other invisible aggregation bands with different sizes could exist in the gel. If Tpi1 constitutes aggregation with many different proteins, it is no wonder even if no visible aggregation band is observed. We thus performed Western blotting using Tpi1-specific antibodies to detect invisible aggregation bands upon total protein staining, and confirmed the existence of many aggregation bands that were disappeared by the removal of cysteines from Tpi1 or the DTT treatment (Figure 8A).

We added these results in Figure 7—figure supplement 1 and 2, and Figure 8 and added relevant descriptions in Results and Discussion as follows:

In Results:

“We confirmed that cysteines were responsible for creating these bands because they were disappeared when cysteine residues were removed from Pgk1 and Eno2 (Figure 7—figure supplementary 1).”

“This aggregation seemed not to affect their expression limits because expression limits of wild-types and cysteine-less mutants (Eno2-C248S and Pgk1-C98S) were indistinguishable (Figure 7—figure supplementary 2). […] To test whether the aggregation restricts the Tpi1 expression limit, we measured expression limits of cysteine-less Tpi1. As shown in Figure 8B, the expression levels of cysteine-less Tpi1 significantly increased.”

In Discussion:

“Overexpression of Eno2, Pgk1, and Tpi1 triggered S–S-bond-connected aggregation (Figure 7 and 8).”

“The deleterious effect of this aggregation, however, seems protein-specific because expression limits of Pgk1 and Eno1 were among highest (Figure 2A), and removal of their cysteine did not increase expression limits of them (Figure 7—figure supplement 1).”

3) Overall the authors play a bit fast and loose with their statistics. First, p values should be accurately reported. Don't write "p<0.05", write "p=0.032". Second, whenever p values are stated it should also be stated what test was used (see e.g. subsection “Mutations in catalytic centers not affecting expression limits of most glycolytic proteins”). Third, correlations should be reported with p values (e.g. subsection “Metabolic perturbations triggered upon overexpression of glycolytic proteins”, last paragraph).

We added statistics as the reviewer suggested. We initially used Student’s *t*-test to calculate *p*-value for most of our measurements, but re-calculated *p*-values using Welch’s *t*-test for more strict significance evaluations. All *p-*values are listed in each figure source data.

4) While the authors describe an interesting system to estimate the limits of protein expression within a cell, there are several discrepancies between vector copy number and measured expression levels, which raises the concern the results can be reflective of the technical experimental setting instead of true limitations.

We answered to this comment in the following specific examples.

In addition, the authors use GFP, an exogenous protein with a high expression, as control. Endogenous proteins, preferably unidirectional and bidirectional enzymes, that show high and low expression levels, will make for better controls to the glycolytic enzymes.

As we mentioned in Introduction, we previously found that GFP can be overexpressed up to the protein burden level which is 15% of the total cellular protein (Kintaka et al., 2016). As far as we recognize, there was no report that any endogenous or exogenous protein is expressed up to this level in *S. cerevisiae*. In other words, we did not know any endogenous protein whose expression limit is this high, and it was impossible to use an endogenous protein as a control in this study. Therefore, we can instead consider this study as the first identification of endogenous unidirectional and bidirectional enzymes that show high and low expression levels that can be used for future research.

The following are specific examples of such discrepancies:A) In Figure 1B and C, why is the maximum growth rate between high and low copy number vector control so different?

We are sorry that our experimental system used here is a bit unusual and difficult to understand. The detailed explanation of this method is omitted because it is previously published (Moriya et al., 2006 and 2012). In this experiment, we used pTOW40836 who carries *URA3* and *leu2-89 (LEU2* with a truncated promoter) as the selection markers. Level of the Ura3 protein expressed from the plasmid is sufficient to fully recover the growth of the host strain (*ura3* and *leu2* deletion) in –uracil (+leucine) conditions even if the plasmid copy number is low because *URA3* has a full strength promoter. In contrast, levels of the Leu2 protein is insufficient to recover the growth in –leucine conditions if the plasmid copy number is low because *leu2-89* has a large deletion in the promoter. This becomes a bias to increase the plasmid copy number up to 150 copies per cell in –leucine conditions (to be more precise, a bias to select cells with higher plasmid copies). However, even in this condition, the growth rate is not fully recovered compared to +leucine conditions probably because the level of Leu2 protein is not sufficient enough.

We added a description below in Results:

“In this experimental system, maximum growth rates of the cells with the vector in +leucine conditions is much higher than those in –leucine conditions (see Figures 1B–C), probably because the copy number of *leu2-89* is not sufficient to fully support the leucine requirement in –leucine conditions.”

B) Subsection “Measurement of expression limits of glycolytic proteins”, second paragraph: The two explanations mentioned by the authors are not mutually exclusive. The authors argue that proteins with low expression and low copy number are harmful for the cell and the ones with low expression and high copy number are repressed due to their high copy number. This is a circular argument and doesn't explain why the copy numbers are high in the first place.

We agree those two explanations are not mutually exclusive. We thus remove ‘independent’ from the sentence. For the latter part, we are sorry for our confusing descriptions; we used ‘because’ not for the mechanistic explanation but the evidence. To avoid confusion, we added exact citations to the data points on the graph in the sentences as follows:

“There could be two reasons that the expression level of a protein is low: (i) its strong overexpression is harmful to cellular growth and (ii) its expression is repressed. […] In contrast, the expression of Glk1, Pyk2, and Pdc6 seemed to be repressed because their copy numbers were higher than the others’ (blue circles in Figure 2D).”

Finally, is the Pearson correlation of 0.3 significant? The authors have dismissed it but they need to show that it is statistically not significant.

We added p value (0.11) that was not significant.

C) The authors claim that 15 percent of the total cellular protein is the limit for overexpression of protein. However, the authors do not observe any correlation between molecular weight and protein expression levels. How do the authors explain this lack of correlation?

In this study, we calculate the unit for the expression level of a protein (AU) from the total intensity of fluorescently-stained protein band, which reflects the total amino acid number within the band (not the molecule number of the protein). Therefore, if two proteins give the same expression AUs, the molecule number in the band of the larger protein is lower as the reviewer claims. For example, Gpm1 and Pgk1 had their expression levels 2.63 AU and 2.27 AU. When those units are normalized with their lengths (248 a.a. and 415 a.a.), their expression levels are 0.010 AU/a.a and 0.005 AU/a.a., so the larger protein expresses lower in the molecule number.

We can think the reason why we did not see any correlation between the expression levels (AU) and lengths is that the cost for protein production but not the number of the protein molecule determines the expression limit. Fifteen percent of the total cellular protein thus means ‘15% of total amino acid incorporated into cellular protein’, but not ‘15% molecules of total protein molecules’. We also want to emphasize that a protein can be expressed up to 15% of total protein only if overexpression of the protein does not cause additional harmful effect than the protein burden. So the expression limit of a harmful protein should be determined independently by the size.

We added these explanations in Results and Discussion.

In Results:

“The AU is considered to reflect the total amino acid number within the band, and the relative number of the protein molecule can be estimated by dividing AU by the protein length. When two proteins with different sizes give the same AUs, the molecule number of the larger protein in the band should be lower.”

In Discussion:

“Among the glycolytic proteins studied here, Pgk1, Gpm1, and Eno2 gave highest expression limits. […] It also suggests that their expression limits are not determined by the molar concentration of the proteins but by the cost of the protein production.”

5) The results depend on how exactly one defines "growth defect" and how accurately one measures it. The paper does not discuss this issue. "growth defect" needs to be defined precisely, and the authors also need to argue that they can measure it with sufficient accuracy. One way by which one could get the result that there are no growth defects even at high levels of overexpression is by using a very insensitive assay.

We define growth defect based on the significance in the difference of maximum growth rates (MGRs) between the cells overexpressing the target protein and the cells with the control vector. We also use the copy numbers of plasmids expressing target proteins as indicators for growth defects upon overexpression of target proteins. It is not easy to argue how accurate our measurements are, because we did not compare our measurements with other more sensitive measurements like competitive growth assay. However, these measurements are sufficiently reproducible to argue growth defects with statistical significance. We observed significant reductions of MGRs of the cells overexpressing most of the glycolytic proteins (*p* < 0.01, Welch’s *t*-test, Figure 1B). Plasmid copy numbers seemed more sensitive indicators for growth defects because all of them overexpressing glycolytic proteins but Pyk2 were less than half of that of the vector control with larger statistical significance (*p* < 0.001, Welch’s *t*-test, Figure 1D). We thus think our experimental system is sufficiently sensitive to discuss growth defects triggered by the overexpression of glycolytic proteins.

We added the explanation for our definition of the growth defect in Materials and methods as follows:

“The maximum growth rate (MGR) was calculated as described previously (Moriya et al., 2006). Average values, SD, and *p*-values of Welch's *t-*test were calculated from biological triplicates. We define growth defect based on the significance in the reduction of maximum growth rate of the cells overexpressing a target protein compared with that with the control vector (*p* < 0.01, Welch’s t-test).”

6) A simple summary table presenting the expression limit and proposed mechanism of toxicity (or not) and the evidence for this for all 29 proteins would be very helpful. This could replace some of the information in Table 1 which could be moved to the supplement.

We are thankful for the reviewer’s suggestion. We made a summary table (Supplementary file 1).

7) In places it is not entirely clear why the authors are only performing mechanistic experiments on a specific subset of the proteins. Again a summary table might help to better communicate what has been tested for which proteins and why.

We made a summary table (Supplementary file 1).

8) 'Repression' implies an active mechanism to lower protein concentration whereas it is just that these proteins are not using optimised codons that increase translation like the other enzymes. It is better to avoid this word and simply talk about 'lower expression'.

In the earlier part of Results, we used ‘repress’ to imply an active mechanism to lower protein concentration because we did not know the mechanism to lower the expression of Glk1, Pyk2, and Pdc6. The mechanism was later turned out to be low codon-optimality. Although we think that lowering codon optimality (deoptimizing) could be an evolutionally active mechanism to lower protein concentration, we agree this word is confusing. We thus changed the description in Discussion as follows:

“The translational rate of some glycolytic proteins, including Glk1, seemed low due to their lower codon optimality (Figure 6).”

9) The metabolic profiling is rather inconclusive. Is the conclusion simply that changes in the quantified metabolites cannot be causing the growth defect? There also doesn't seem to be much of a connection between the results of the computational simulations and the metabolic profiling, so it's not at all clear how useful the simulations are.

We agree that the metabolic profiling is inconclusive. A big issue we have recognized through the metabolic analysis was we hardly know how much change of which metabolite triggers growth defect. For example, does a threefold change in the F16bP level cause growth defect? As far as we know, this issue has never been assessed. Moreover, we also have recognized that we currently do not have any systematic way to identify metabolic changes directly triggered by the overexpression of an enzyme because metabolism is interconnected and overexpression of a protein could cause non-specific perturbations that ultimately affect metabolism. To answer this issue, we have to wait until a large-scale survey to connect metabolic changes and growth defects, preferably with systematic perturbations such as deletion and overexpression is performed. As described above, we observed a threefold difference between wild-type and CC mutant Pfk2 (Figure 5A). However, we cannot conclude this causes the difference in their expression limits with our current knowledge. We thus do not want to simply conclude that changes in the quantified metabolites cannot be causing the growth defects.

Above reasons are why we used a mathematical model as a reference. Using a mathematical model, we can predict potential metabolic changes triggered by overexpression of an enzyme without considering other unknown effects than its metabolic activity. As in the case of Pfks and Hxks overexpression in the simulation (Figures 4B, D), we can predict the pattern of metabolic changes. We again cannot define how much changes in these metabolites could trigger growth defects in the simulation (and in vivo). However, if the metabolic changes are divergent and almost catastrophic as observed in the Pfks and Hxks simulations (~1,000 fold increase in some metabolites, Figures 4B, D), we could claim metabolic changes triggered by the overexpression of these enzymes should cause growth defects. Hence, we expected to obtain similar metabolic changes upon overexpression of Pfks whose catalytic activities seemed to trigger growth defects. However, we did not observe such a big changes, and the pattern was not consistent (Figures 5, Figure 5—figure supplement 1). We also agree that the usefulness of the simulations is unclear with these results, but we at least could show one approach to attack above uncertain biological issue.

We agree that this is a critical argument need to be included in Discussion and described as follows:

“Through the metabolic analysis, we realized that we currently do not have any systematic way to identify metabolic changes directly triggered by the overexpression of an enzyme, because metabolism is interconnected and overexpression of a protein could cause non-specific perturbations that ultimately affect metabolism. […] To precisely answer these issues, we need a much deeper understanding of the connection between metabolite levels and the cellular growth.”

10) There is an obvious other source of potential growth defects that have been discussed widely in the literature but that aren't mentioned at all: Toxic effects due to protein misfolding or misinteractions. For example, Geiler-Samerotte et al. measured the effect of overexpressed, misfolded GFP on yeast growth and found an effect in proportion to the amount of misfolded protein (https://doi.org/10.1073/pnas.1017570108). Also concentration-dependent liquid demixing e.g. Bolognesi et al., 2016. Similar topics have been discussed in the literature for a long time, see e.g. this review: https://www.nature.com/articles/nrg2662. No additional work required, but discussion is needed.

We thank the reviewer for noticing us a critical argument. We had intended to include these mechanisms as promiscuous interaction in Introduction, but we recognized that it was not sufficient and needed to discuss them more.

We thus added the following descriptions in Discussion:

“Protein misfolding or misinteraction is considered to cause toxicity upon high-level expression of a protein with low translational robustness, low folding stability, or a high propensity for misinteraction (Drummond and Wilke, 2009; Zhang and Yang, 2015). […] We do not think this mechanism caused growth defects upon overexpression of glycolytic proteins studied here because they are less structurally-disordered (Moriya, 2015) and not nucleic-acid-binding proteins.”